# Pathway dynamics can delineate the sources of transcriptional noise in gene expression

**Lucy Ham[1]\*, Marcel Jackson[2], Michael PH Stumpf[3]\***

[1]School of BioSciences, University of Melbourne, Melbourne, Australia; [2]Department of Mathematics and Statistics, La Trobe University, Melbourne, Australia; [3]School of Mathematics and Statistics, University of Melbourne, Melbourne, Australia

**Abstract** Single-cell expression profiling opens up new vistas on cellular processes. Extensive cell-to-cell variability at the transcriptomic and proteomic level has been one of the stand-out observations. Because most experimental analyses are destructive we only have access to snapshot data of cellular states. This loss of temporal information presents significant challenges for inferring dynamics, as well as causes of cell-to-cell variability. In particular, we typically cannot separate dynamic variability from within cells ('intrinsic noise') from variability across the population ('extrinsic noise'). Here, we make this non-identifiability mathematically precise, allowing us to identify new experimental set-ups that can assist in resolving this non-identifiability. We show that multiple generic reporters from the same biochemical pathways (e.g. mRNA and protein) can infer magnitudes of intrinsic and extrinsic transcriptional noise, identifying sources of heterogeneity. Stochastic simulations support our theory, and demonstrate that 'pathway-reporters' compare favourably to the well-known, but often difficult to implement, dual-reporter method.

**\*For correspondence:**
lucy.ham@unimelb.edu.au (LH);
mstumpf@unimelb.edu.au (MPHS)

**Competing interests:** The authors declare that no competing interests exist.

**Reviewing editor:** Ramon Grima,

## Introduction

Noise is a fundamental aspect of every cellular process *Shahrezaei and Swain, 2008b*. Frequently it is even of functional importance, for example in driving cell-fate transitions. Sometimes it can afford evolutionary advantages, for example, in the context of bet-hedging strategies. Sometimes, it can be a nuisance, for example, when it makes cellular signal processing more difficult. But noise is nearly ubiquitous at the molecular scale, and its presence has profoundly shaped cellular life. Analysing and understanding the sources of noise, how it is propagated, amplified or attenuated, and how it can be controlled, has therefore become a cornerstone of modern molecular cell biology.

Noise arising in gene expression has arguably attracted most of the attention so far (but see e.g. *Filippi et al., 2016* and *Jetka et al., 2018* for the analysis of noise at the signalling level). Generally speaking, gene expression noise is separable into two sources of variability, as pioneered by *Swain et al., 2002*. *Intrinsic noise* is generated by the dynamics of the gene expression process itself. The process, however, is often influenced by other external factors, such as the availability of promoters and of RNA polymerase, the influence of long noncoding RNA as a transcriptional regulator *Goodrich and Kugel, 2006*, as well as differences in the cellular environment. Such sources of variability contribute *extrinsic noise*, and reflect the variation in gene expression and transcription activity across the cell population. As such, understanding extrinsic noise lies at the heart of understanding cell-population heterogeneity.

So far, identifying the sources of gene expression noise from transcriptomic measurements alone has proven difficult *Paulsson, 2004*; *Pedraza and Paulsson, 2008*. The fundamental hindrance lies in the fact that single-cell RNA sequencing, which provides most of the available data, is destructive, so that datasets reflect samples from across a population, rather than samples taken repeatedly

**eLife digest** In biology, seemingly random variation within or between cells can have significant effects on a number of cellular processes, like how cells divide and develop. For example, how often a gene is switched on, or 'expressed', can randomly fluctuate over time. This 'noise' may lead to a cell having slightly more of a particular molecule, causing it to behave differently to other cells in the population. However, it is currently unclear how this random variation is created and controlled in cells, and what effect this has on biological systems as a whole.

When a gene is expressed, its sequence typically gets copied in to a molecule called mRNA, which is then processed and used to build the protein encoded by the gene. By measuring the levels of mRNA molecules in individual cells, researchers have been able to investigate how gene expression varies within populations. These experiments are carried out on dead cells at a single point in time, and mathematical models are then applied to detect noise in the molecular data.

This approach, however, precludes how noise changes over time, making it difficult to determine the source of cell-to-cell variability. In particular, whether the variation detected is the result of genuine random molecular changes (intrinsic noise), or external factors – such as temperature and pH – fluctuating in the cells environment (extrinsic noise).

Here, Ham et al. have built on previous mathematical models to identify a new approach for investigating the source of molecular noise. They found that for any given gene it is impossible to understand what causes its activity levels to vary just from data on its mRNA levels. Instead, information on other molecules that are affected by expression of the gene (termed 'pathway reporters') can provide a clearer picture of whether molecular variability is the result of intrinsic or extrinsic noise.

The mathematical models developed by Ham et al. reveal what can and cannot be learned about noise from gene expression data. Furthermore, pathway-reporters are easier to measure experimentally than other reporters that are typically used to study the origins and effects of cell-to-cell variability. These findings could help researchers design single cell experiments that are better for studying noise, leading to a deeper understanding of how different types of variation impact cell biology.

from the same cell. As temporal information is lost in such measurements *Komorowski et al., 2011*, it may be impossible to distinguish temporal variability within individual cells (e.g. burstiness), from ensemble variability across the population (i.e. extrinsic noise). A number of numerical and experimental studies have suggested this confounding effect *Jones and Elf, 2018*; *Jones et al., 2014*; *Zopf et al., 2013*, showing that systems with intrinsic noise alone exhibit behaviour that is indistinguishable from systems with both extrinsic and intrinsic noise. This is examined more formally in *Ham et al., 2020a*, where we show that the moment scaling behaviour and transcript distribution may be indistinguishable from situations with purely intrinsic noise. The limitations in inferring dynamics from population data are becoming increasingly evident, and a number of studies that seek to address some of these problems have emerged *Skinner et al., 2016*; *Gorin and Pachter, 2020a*.

Here we provide a detailed analysis of the extent to which sources of variability are identifiable from population single-cell omics data. We are able to prove rigorously that it is in general impossible to identify the sources of variability, and consequently, the underlying transcription dynamics, from observed transcript abundance distributions alone. Systems with intrinsic noise alone can always present identically to similar systems with extrinsic noise. For practical purposes, the effect does not appear to depend on the precise choice of distribution, but holds more generally. Moreover, we demonstrate that extrinsic noise invariably distorts the apparent degree of burstiness of the underlying system: data which seems 'bursty' is not necessarily generated by a bursty process, if there is cell-to-cell variability across the population. This is a stronger non-identifiability result than has previously been obtained *Ingram et al., 2008*; *Shahrezaei and Swain, 2008a*; *Khammash, 2009*; *Munsky et al., 2009*; *Komorowski et al., 2011*, and has important ramifications for our analysis of experimental data: we cannot assess causes of transcriptional variability, if we do not simultaneously assess cell-to-cell variability in the transcriptional machinery. Our results highlight (in fact prove

mathematically) the requirement for additional information, beyond the observed copy number distribution, in order to constrain the space of possible dynamics that could give rise to the same distribution.

This seemingly intractable problem can at least partially be resolved with a brilliantly simple approach: the dual-reporter method *Swain et al., 2002*. In this approach, noise can be separated into extrinsic and intrinsic components, by observing correlations between the expression of two independent, but identically distributed fluorescent reporter genes. Dual-reporter assays have been employed experimentally to study the noise contributed by both global and gene-specific effects *Elowitz et al., 2002*; *Raser and O'Shea, 2005*; *Raj et al., 2006*. A particular challenge, however, is that dual reporters are rarely identically regulated *Raj et al., 2006*; *Quarton et al., 2020*, and are not straightforward to set up in every system. More recently, it has been shown that the independence of dual reporters cannot be guaranteed in some systems *Naik et al., 2021*. As a result, there have been efforts in developing alternative methods for decomposing noise *Quarton et al., 2020*; *Singh, 2014*; *Lin and Amir, 2021*. Here we develop a widely applicable generalisation (and simplification) of the original dual-reporter approach *Swain et al., 2002*. We demonstrate that non-identical and not-necessarily independent reporters can provide an analogous noise decomposition. Our result shows that measurements taken from the same biochemical pathway (e.g. mRNA and protein) can replace dual reporters, enabling the noise decomposition to be obtained from a single gene. This completely circumvents the requirement of conditionally independent and identically regulated reporter genes. The results obtained from our 'pathway-reporter' method are also borne out by stochastic simulations, and compare favourably with the dual-reporter method. In the case of constitutive expression, the results obtained from our decomposition are identical to those obtained from dual reporters. For bursty systems, we show that our approach provides a satisfactorily close approximation, except in extreme cases. Our methodology is verified mathematically for the most common models of gene transcription, but holds independently of the specific nature of the gene expression dynamics, as we verify in silico across a range of more detailed models.

## Materials and methods

A simple model for stochastic mRNA dynamics is the *Telegraph model*: a two-state model describing promoter switching, transcription, and mRNA degradation. In this model, all parameters are fixed, and gene expression variability arises due to the inherent stochasticity of the transcription process. As discussed above, this process will often be influenced by extrinsic sources of variability, and so modifications to account for this additional source of variability are required.

### The telegraph model

The Telegraph model was first introduced in *Ko, 1991*, and has since then been widely employed in the literature to model bursty gene expression in eukaryotic cells *Bahar Halpern et al., 2015*; *So et al., 2011*; *Suter et al., 2011*; *Larsson et al., 2019*. In this model, the gene switches probabilistically between an active state and an inactive state, at rates $\lambda$ (on-rate) and $\mu$ (off-rate), respectively. In the active state, mRNAs are synthesised according to a Poisson process with rate $K$, while in the inactive state, transcription does either not occur, or possibly occurs at some lower Poisson rate, $K_0 \ll K$. Degradation of mRNA molecules occurs independently of the gene promoter state at rate $\delta$. *Figure 1A* shows a schematic of the Telegraph model. Throughout the discussion here, we will rescale all parameters of the Telegraph model by the mRNA degradation rate, so that $\delta = 1$. The steady-state distribution for the mRNA copy number can be explicitly calculated as *Peccoud and Ycart, 1995*,

$$\tilde{p}_T(n;\theta) = \frac{K^n \lambda^{(n)}}{n!(\mu+\lambda)^{(n)}} {}_1F_1(\lambda+n, \lambda+\mu+n, -K).$$

(1)

Here, $\theta$ denotes the parameter vector $(\mu, \lambda, K, \delta)$, the function ${}_1F_1$ is the confluent hypergeometric function *Abramowitz and Stegun, 1965*, and, for real number $x$ and positive integer $n$, the notation $x^{(n)}$ abbreviates the rising factorial of $x$ (also known as the Pochhammer function). Throughout, we refer to the probability mass function $\tilde{p}_T(n;\theta)$ as the *Telegraph distribution with parameters* $\theta$.

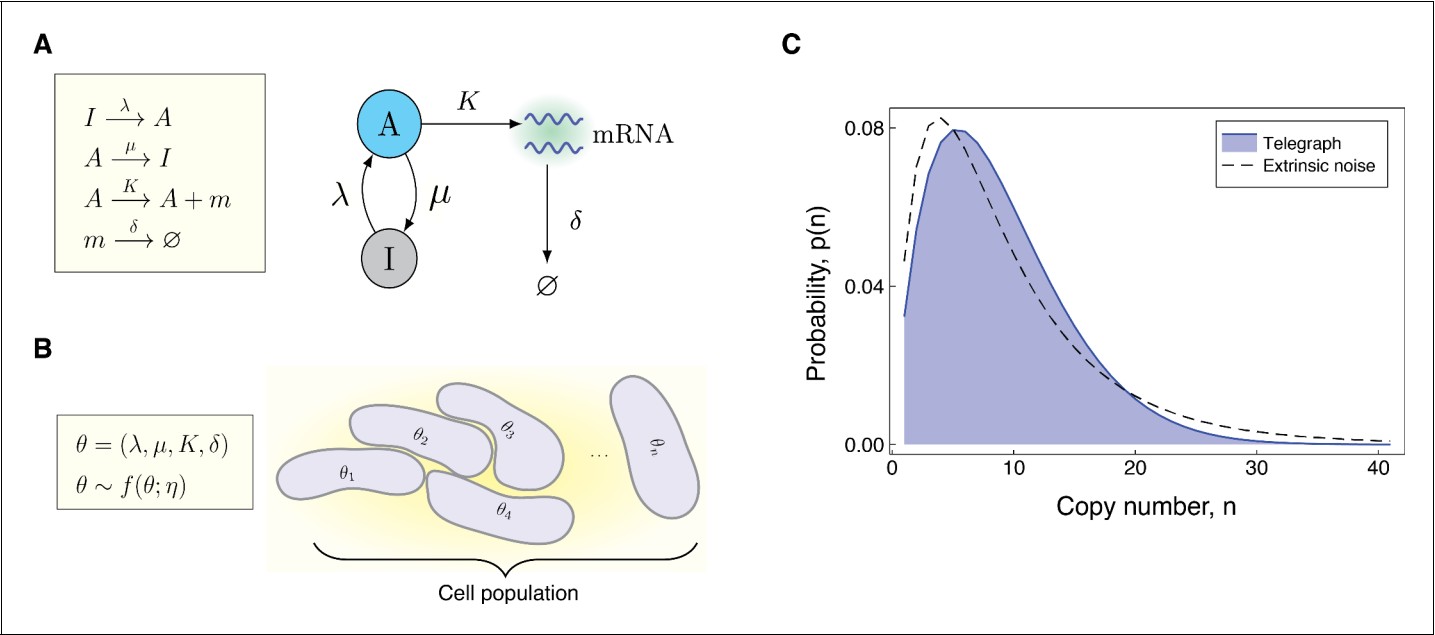

**Figure 1.** Modeling the effects of both intrinsic and extrinsic noise. (**A**) A schematic of the Telegraph process, with nodes *A* (active) and *I* (inactive) representing the state of the gene. Transitions between the states *A* and *I* occur stochastically at rates μ and λ, respectively. The parameter *K* is the mRNA transcription rate, and δ is the degradation rate. (**B**) The compound model incorporates extrinsic noise by assuming that parameters θ of the Telegraph model vary across an ensemble of cells, according to some probability distribution $f(\theta; \eta)$. (**C**) Variation in the parameters across the cell population leads to greater variability in the mRNA copy number distribution.

*Constitutive* gene expression is a limiting case of the Telegraph model, which arises when the off-rate μ is 0, so that the gene remains permanently in the active state. In this case, the Telegraph distribution simplifies to a Poisson distribution with rate *K*; the distribution $\mathrm{Pois}(K)$.

At the opposite extreme is *instantaneously bursty* gene expression in which mRNA are produced in very short bursts with potentially prolonged periods of inactivity in between. This mode of gene expression has been frequently reported experimentally, particularly in mammalian genes *Raj et al., 2006*; *Bahar Halpern et al., 2015*; *Suter et al., 2011*; *Larsson et al., 2019*. Transcriptional bursting may be treated as a limit of the Telegraph model, where the off-rate, μ, tends to infinity, while the on-rate λ remains fixed. In this limit, it can be shown *Jones and Elf, 2018*; *Ham et al., 2020a* that the Telegraph distribution converges to the negative binomial distribution $\mathrm{NegBin}(\lambda, \frac{K}{\mu+K})$.

## The compound distribution

We can account for random cell-to-cell variation across a population by way of a compound distribution *Ham et al., 2020b*

$$\tilde{q}(n; \eta) = \int \tilde{p}(n; \theta) f(\theta; \eta) \, d\theta, \tag{2}$$

where $\tilde{p}(n; \theta)$ is the stationary probability distribution of a system with fixed parameters θ and $f(\theta; \eta)$ denotes the multivariate distribution for θ with hyperparameters η. Often we will take $\tilde{p}(n; \theta)$ to be the stationary probability distribution of the Telegraph model ((1)), and refer to (2) as the *compound Telegraph distribution*. Sometimes $\tilde{p}(n; \theta)$ will be the Poisson distribution or the negative binomial distribution, depending on the underlying mode of gene activity. *Figure 1B* gives a pictorial representation of the compound distribution.

The compound distribution is valid in the case of ensemble heterogeneity, that is, when parameter values differ between individual cells according to the distribution $f(\theta; \eta)$, but remain constant over time *Filippi et al., 2016*. This model is also a valid approximation for individual cells with dynamic parameters, provided these change sufficiently slower than the transcriptional dynamics *Lenive et al., 2016*. In general, the compound Telegraph distribution $\tilde{q}(n; \eta)$ will be more dispersed

than a Telegraph distribution to account for the uncertainty in the parameters; see *Figure 1C*. Such dispersion is widely observed experimentally, and as demonstrated in *Ham et al., 2020a*, reflects the presence of extrinsic noise.

In the context of gene expression, it has been shown experimentally that some of the primary sources of extrinsic noise have an autocorrelation time comparable to the cell cycle *Rosenfeld et al., 2005*. It is these slow changes in variability that justify the assumptions of the compound model. Typical sources of extrinsic variability for each parameter in the Telegraph model are summarised in *Table 1* of *Ham et al., 2020a*. A further significant source of heterogeneity arises from the differences in cell-cycle phases across the population *Skinner et al., 2016*. Such effects have been shown to obscure the precise underlying transcriptional dynamics *Zopf et al., 2013*; *Huh and Paulsson, 2011*, and impede the inference of transcriptional parameters from experimental data *Beentjes et al., 2020*. The compound model we consider here is able to capture this variability, provided that the parameter change within each cell phase is relatively slow, and any dynamic parameter changes during the transition between phases can be considered as ephemeral. A more explicit treatment of the mechanisms and changes during the cell cycle is challenging to study analytically, and theoretical modelling is only in its infancy *Cao and Grima, 2020*; *Beentjes et al., 2020*; *Perez-Carrasco et al., 2020*. Later, we verify our proposed noise decomposition on detailed models of gene transcription, incorporating salient features of the cell-cycle, such as gene replication, dosage compensation, binomial partitioning of products due to cell division, and cell-cycle length variability.

## Results

### Identifiability considerations

Decoupling the effects of extrinsic noise from experimental measurements has been notoriously challenging. In the context of (2), the distribution $f(\theta; \eta)$ reflects the population heterogeneity, but experimental data provides samples only of $\tilde{q}(n; \eta)$. How much can we deduce of the underlying dynamics (that is, $\tilde{p}(n; \theta)$), and the population heterogeneity ($f(\theta; \eta)$), from measurements of transcripts from across the cell population ($\tilde{q}(n; \eta)$)?

Of course, even though we may be able to infer the parameter $\eta$ from experimental data, the expression $\tilde{p}(n; \theta)$ is really a family of distributions, parameterised by $\theta$. This presents two fundamental challenges. The first is the possibility that there are different families of distributions $\tilde{p}(n; \theta)$ that can yield the same compound distribution, $\tilde{q}(n; \eta)$, but which are generated by different mechanisms, that is noise distributions, $f(\theta; \eta)$. The second, perhaps more subtle, challenge is that, even for a fixed family of distributions $\tilde{p}(n; \theta)$ it may be possible that different choices of the noise distribution $f(\theta; \eta)$ could still yield the same compound distribution $\tilde{q}(n; \eta)$. In these situations, we cannot hope to infer a unique solution for the noise distribution. This belongs to the important class of *identifiability problems* Villaverde, *Villaverde and Banga, 2017*; it has important ramifications for the

**Table 1.** Summary of the non-identifiability results.

in lines 1, 3, and 5 are our contributions, while the remaining representations (lines 2 and 4) are known and can be obtained as special cases of our results. Note that here we use $\mathrm{Tele}(\lambda, \mu, K)$ to denote a Telegraph distribution with parameters $\lambda, \mu, K$. In lines 3 and 4, the parameter $\beta > 0$ can be chosen freely and determines the mean burst intensity in the resulting compound system. In line 5, the parameters $b, \theta > 0$ are again mean burst intensities, and $b$ can be chosen freely in the determination of the distribution of $\theta$.

| Copy no. distribution $\tilde{q}(n; \eta)$ | Underlying distribution $\tilde{p}(n; \theta)$ | Noise distribution $f(\theta; \eta)$ |
|---|---|---|
| $\mathrm{Tele}(\lambda, \mu', K')$ | $\mathrm{Tele}(\lambda, \mu, K)$ | $K \sim \mathrm{Beta}_{K'}(\lambda + \mu, \mu' - \mu)$ |
| $\mathrm{Tele}(\lambda, \mu', K')$ | $\mathrm{Pois}(K)$ | $K \sim \mathrm{Beta}_{K'}(\lambda, \mu')$ |
| $\mathrm{NegBin}(\lambda, \frac{\beta}{\beta+1})$ | $\mathrm{Tele}(\lambda, \mu, K)$ | $K \sim \mathrm{Gamma}(\lambda + \mu, \beta)$ |
| $\mathrm{NegBin}(\lambda, \frac{\beta}{\beta+1})$ | $\mathrm{Pois}(K)$ | $K \sim \mathrm{Gamma}(\lambda, \beta)$ |
| $\mathrm{NegBin}(\lambda', \frac{b}{b+1})$ | $\mathrm{NegBin}(\lambda, \frac{\theta}{\theta+1})$ | $\theta \sim \mathrm{BetaPrime}_b(\lambda - \lambda', \lambda')$ |

interpretability of parameter estimates obtained from experimental data *Ingram et al., 2008*. Indeed, if two or more model parameterisations are observationally equivalent (in this case, in the form of the transcript abundance distribution $\tilde{q}(n;\eta)$), then not only does this cast doubt upon the suitability of the model to represent (and subsequently predict) the system, but it also obstructs our ability to infer mechanistic insight from experimental data.

An example of the first identifiability problem arises from a well-known example of a compound distribution, (2): when $f(\theta;\eta)$ is a gamma distribution and $\tilde{p}(n;\theta)$ is a Poisson distribution, corresponding to constitutive gene expression, the resulting compound distribution $\tilde{q}(n;\eta)$ is a negative binomial distribution *Greenwood and Yule, 1920*. But this is the same distribution as that arising from instantaneously bursty gene expression *Ham et al., 2020a*. Such identifiability instances may be circumvented if there is confidence in the basic mode of gene activity, that is, if there is reason to believe that a gene is not constitutively active, for example. We find, however, that there are numerous instances that can present insurmountable identifiability problems.

## Bursty gene expression

We first observe that any Telegraph distribution with fixed parameters can be identically obtained from a Telegraph distribution with parameter variation. As shown in the supplementary material (Appendix Pathway dynamics can delineate the sources of transcriptional noise in gene expression), any Telegraph distribution $\tilde{p}_T(n;\lambda,\mu',K')$ can be written as,

$$\tilde{p}_T(n;\lambda,\mu',K') = \int_0^{K'} \tilde{p}_T(n;\lambda,\mu,t)f_{K'}(t;\lambda+\mu,\mu'-\mu)\,dt, \tag{3}$$

where $\mu<\mu'$ and $f_{K'}(t;\lambda+\mu,\mu'-\mu)$ is the probability density function for a scaled beta distribution $\mathrm{Beta}_K(\lambda+\mu,\mu'-\mu)$ with support $[0,K']$. Thus, any Telegraph distribution can be obtained by varying the transcription rate parameter on a narrower Telegraph distribution (i.e. with a smaller off-rate) according to a scaled beta distribution. *Figure 2A* (top panel) compares the representation obtained in (3) with the corresponding fixed-parameter Telegraph distribution for two different sets of parameters. When $\mu=0$ the representation given in (3) simplifies to the well-known Poisson representation of the Telegraph distribution in terms of the scaled beta distribution *Srividya et al., 2009*.

## Instantaneously bursty gene expression

The previous result extends to instantaneously bursty systems. The copy number distribution of an instantaneously bursty system can be obtained from both bursty and instantaneously bursty dynamics, provided that there is appropriate parameter variation. The supplementary material contains the relevant derivations; refer to Appendix Pathway dynamics can delineate the sources of transcriptional noise in gene expression. In the following, we let $\tilde{p}_{\mathrm{NB}}(n;r,\beta)$ denote the probability mass function of a $\mathrm{NegBin}(r,\frac{\beta}{\beta+1})$ distribution, where $\beta>0$. Then for any negative binomial distribution of the form $\mathrm{NegBin}(\lambda,\frac{\beta}{\beta+1})$ we have,

$$\tilde{p}_{\mathrm{NB}}(n;\lambda,\beta) = \int_0^{\infty} \tilde{p}_T(n;\lambda,\mu,t)f(t;\lambda+\mu,\beta)\,dt, \tag{4}$$

where $f(t;\lambda+\mu,\beta)$ is the probability density function of a $\mathrm{Gamma}(\lambda+\mu,\beta)$ distribution. This result generalises the aforementioned well-known representation of the negative binomial distribution *Greenwood and Yule, 1920*, which corresponds to the particular case of $\mu=0$. In *Figure 2A* (middle panel), we compare the representation obtained in (4) with the corresponding fixed-parameter negative binomial distribution for two different sets of parameters.

We also obtain the following representation for a negative binomial distribution in terms of a scaled beta prime distribution,

$$\tilde{p}_{\mathrm{NB}}(n;\lambda',b) = \int_{\frac{1}{b}}^{\infty} \tilde{p}_{\mathrm{NB}}(n;\lambda,\theta)f_b(b\theta-1;\lambda-\lambda',\lambda')\,d\theta, \tag{5}$$

where $f_b(b\theta-1;\lambda-\lambda',\lambda')$ is the probability mass function of a scaled beta prime $\mathrm{BetaPrime}_b(\lambda-\lambda',\lambda)$ distribution, where $b>0$ and $\lambda>\lambda'$. This equivalently corresponds to scaled beta noise $\mathrm{Beta}_b(\lambda-\lambda',\lambda')$ on the inverse of the expected burst intensity. Thus, the distribution of any instantaneously bursty

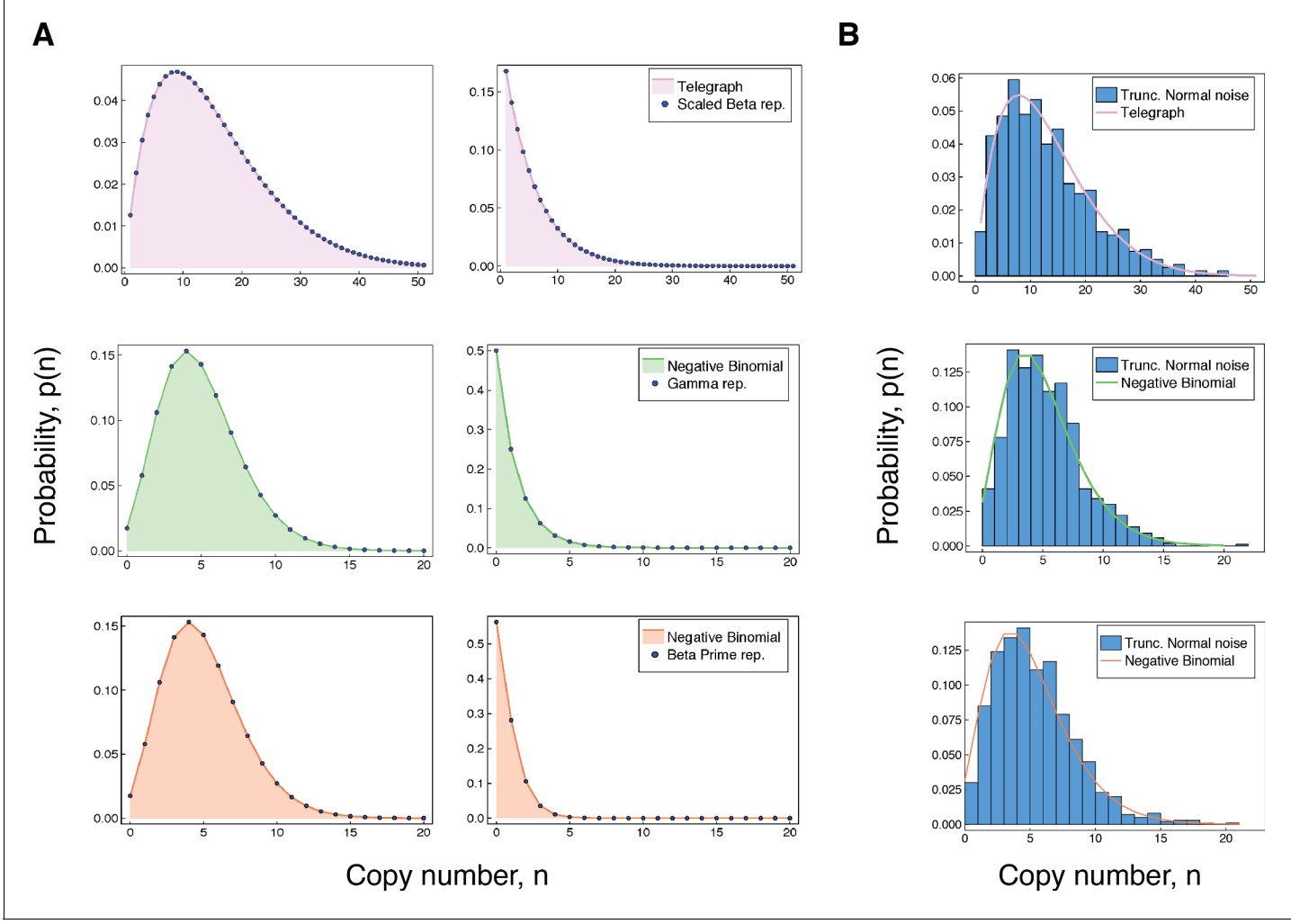

**Figure 2.** Accuracy of our integral representations for the Telegraph and negative binomial distribution. (**A**) For each of the results in (3 - 5), we compare the (fixed-parameter) Telegraph and negative binomial distributions with their respective compound representations for two different sets of parameter values. The top panel (pink) shows comparisons for (3), with parameter values (left) $\lambda = 2$, $\mu' = 12$, $K' = 100$, $\mu = 3$, and $K \sim \mathrm{Beta}_{K'}(5, 9)$, and (right) $\lambda = 1$, $\mu' = 20$, $K' = 100$, $\mu = 2$ and $K \sim \mathrm{Beta}_{K'}(3, 18)$. The middle panel (green) gives comparisons for (4), with parameter values (left) $\lambda = 10$, $\beta = 2$, $\mu = 2$ and $K \sim \mathrm{Gamma}(12, 2)$ and (right) $\lambda = 1$, $\beta = 1$, $\mu = 2$ and $K \sim \mathrm{Gamma}(3, 1)$. The bottom panel (coral) gives comparisons for (5). The parameter values (left) are $\lambda' = 10$, $\lambda = 15$ and $c = 2$ and (right) are $\lambda' = 2$, $\lambda = 5$ and $c = 3$. (**B**) The top figure compares a $\mathrm{Telegraph}(2, 4, 60)$ distribution with samples from a compound Telegraph distribution with normal noise $\mathrm{Norm}(37, 10)$ on the transcription rate parameter. The middle figure compares a $\mathrm{NegBin}(5, 0.5)$ with samples from a compound Telegraph distribution with normal noise $\mathrm{Norm}(5.5, 2.3)$ on the transcription rate parameter. The bottom figure compares a $\mathrm{NegBin}(5, 1)$ distribution with samples from a compound negative binomial distribution with normal noise $\mathrm{Norm}(2.3, 0.6)$ on the burst intensity parameter.

system with mean burst intensity $b$ can be obtained from one with greater burst frequency, by varying the mean burst intensity θ according to a shifted beta prime distribution. *Figure 2A* (bottom panel), compares the representation obtained in (5) with the associated fixed-parameter negative binomial distribution for two different sets of parameters.

## An exception: constitutive expression

It has long been known *Feller, 1943* that a compound Poisson distribution uniquely determines the compounding distribution. In the context of (2), this means the full extrinsic noise distribution $f(\theta, \eta)$ is identifiable from $\tilde{q}(n; \eta)$. As we will demonstrate in related work (Ham et al., in preparation) , it is therefore possible to extract the extrinsic noise distribution, $f(\theta, \eta)$, from transcript copy number measurements.

## Implications for parameter inference

Estimates of kinetic parameters from experimental data suggest that gene expression is often either bursty or instantaneously bursty (i.e., $\mu \gg \lambda$). In turn, the assumption that gene-inactivation events occur far more frequently than gene-activation events is often used to derive other models of stochastic mRNA dynamics *Jia and Grima, 2020*; *Cao and Grima, 2020*; *Beentjes et al., 2020*. The representations given in (3 – 5), however, show that both estimating parameters and the underlying dynamics from the form of the copy number distribution alone can be misleading. Noise on the transcription rate will invariably produce copy number data that is suggestive of a more bursty model. To illustrate this, consider an example in which the underlying process is a (mildly) bursty Telegraph system with distribution $\tilde{p}_T(n; 2, 3, K)$. Now assume that noise on the transcription rate parameter $K$ follows the scaled Beta distribution on the interval $[0, 100]$ with $\alpha = \lambda + \mu = 2 + 3 = 5$ and $\beta = \mu' - \mu = 12 - 3 = 9$. The shape of this noise distribution closely resembles a slightly skewed Gaussian distribution, with the majority of transcription rates between around 10 and 60. This noise on the transcription rate $K$ within the Telegraph system $\tilde{p}_T(n; 2, 3, K)$ will present identically to the significantly burstier system $\tilde{p}_T(n; 2, 12, 100)$.

It is of practical importance to recognise that, while the non-identifiability results (summarised in *Table 1*) are dependent on specific noise distributions, for practical purposes any similar distribution will produce a similar effect. To demonstrate this, we replace the various noise distributions required for the representations in (3 – 5), with suitable normal distributions truncated at 0. In each case, we sample 1000 data points from the corresponding compound distribution, and compared this with the associated fixed-parameter copy number distribution. The results are shown in *Figure 2B*. The truncated normal distribution is not chosen on the basis of biological relevance, but rather to demonstrate that even a symmetric noise distribution (except for truncation at 0) produces qualitatively similar results to the distributions used in the precise non-identifiability results. In every case, the effect of varying the transcription rate or burst intensity parameter according to a unimodal noise distribution is to produce copy number distributions that are generally consistent with systems that appear burstier.

Finally, we note that our results explain previous empirical observations that static measurements of mRNA are not always sufficient to infer the underlying dynamics of gene activity. *Skinner et al., 2016* address some of these limitations by quantifying both nascent and mature mRNA in individual cells, as well incorporating cell-cycle effects into their analysis of two mammalian genes. A more developed treatment of model identifiability is given in *Gorin and Pachter, 2020a*, where it shown how a stochastic model incorporating the downstream processing of mRNA can be used to distinguish a particular instance of non-identifiability. More specifically, the authors consider the non-identifiability problem noted in *Ham et al., 2020a*, arising from the Gamma-Poisson compound representation of the negative binomial distribution; a particular case of (4) above. Despite identical distributions at the nascent level, the marginal distributions of the processed (mature) mRNA are found to be substantially different. It is likely that a similar analysis will be valuable in the context of the other identifiability problems we have given in *Table 1*. In the next section, we take a more general approach to resolving non-identifibiality and exploit the properties of complex gene expression dynamics to determine not only the presence of extrinsic noise, but also estimate its magnitude.

## Resolving non-identifiability

The results of the previous section show that additional information, beyond the observed copy number distribution, is required to constrain the space of possible dynamics that could give rise to the same distribution. One way to narrow this space of possibilities, is to determine the intrinsic and extrinsic contributions to the total variation in the system.

## The dual-reporter method

The total gene expression noise, as measured by the squared coefficient of variation $\eta^2$, can be decomposed exactly into a sum of intrinsic and extrinsic noise contributions *Swain et al., 2002*. The decomposition applies to dynamic noise *Hilfinger and Paulsson, 2011*, and generalises to higher moments in *Hilfinger et al., 2012*. Sets of dual reporters at multiple levels of the transcriptional pathway has been shown to achieve a finer breakdown of noise into subcategories *Bowsher and Swain, 2012*. As shown in *Hilfinger and Paulsson, 2011*, the noise decomposition is equivalent to

the normalised Law of Total Variance (*Ross, 2014*). Indeed, if $X$ is the random variable denoting the number of molecules of a certain species (e.g. mRNA or protein) in a given cell, then we can decompose the total noise by conditioning $X$ on the state $\mathbf{Z} = (Z_1, \ldots, Z_n)$ of the environmental variables $Z_1, \ldots, Z_n$,

$$\eta_X^2 = \frac{\mathrm{E}(\mathrm{Var}(X; \mathbf{Z}))}{\mathrm{E}(X)\mathrm{E}(Y)} + \frac{\mathrm{Var}(\mathrm{E}(X; \mathbf{Z}))}{\mathrm{E}(X)\mathrm{E}(Y)} \equiv \eta_{\mathrm{int}}^2 + \eta_{\mathrm{ext}}^2. \tag{6}$$

It has been shown *Swain et al., 2002*; *Hilfinger and Paulsson, 2011* that if $X_1$ and $X_2$ are random variables denoting the expression levels of independent (conditional on $\mathbf{Z}$) and identically distributed gene reporters, then the extrinsic noise contribution $\eta_{\mathrm{ext}}^2$ in (6) can be identified by the normalised covariance between $X_1$ and $X_2$,

$$\eta_{\mathrm{ext}}^2 = \frac{\mathrm{Cov}(X_1, X_2)}{\mathrm{E}(X_1)\mathrm{E}(X_2)}. \tag{7}$$

## Decomposing noise with non-identical reporters

The dual-reporter method requires distinguishable measurements of transcripts or proteins from two conditionally independent and identically distributed reporter genes integrated into the same cell. In practice, however, dual reporters rarely have identical dynamics, which is widely considered to be a significant challenge to interpreting experimental results *Quarton et al., 2020*. We show that, under certain conditions, the decomposition in (6) can alternatively be obtained from non-identically distributed and not-necessarily independent reporters.

Our result relies on the observation that the covariance of any two variables can be decomposed into the expectation of a conditional covariance and the covariance of two conditional expectations (the Law of Total Covariance *Ross, 2014*). If $X$ and $Y$ denote, for example, the numbers of molecules of two chemical species (eg. mRNA and protein) in a given cell, then the covariance of $X$ and $Y$ can be decomposed by conditioning on the state $\mathbf{Z} = (Z_1, \ldots, Z_n)$ of the environmental variables $Z_1, \ldots, Z_n$,

$$\mathrm{Cov}(X, Y) = \underbrace{\mathrm{E}(\mathrm{Cov}(X, Y; \mathbf{Z}))}_{\text{intrinsic}} + \underbrace{\mathrm{Cov}(\mathrm{E}(X; \mathbf{Z}), \mathrm{E}(Y; \mathbf{Z}))}_{\text{extrinsic}}. \tag{8}$$

We will see that in many cases of interest the random variable $\mathrm{E}(X; \mathbf{Z})$ (as a function of $\mathbf{Z}$) *splits across common variables* with $\mathrm{E}(Y; \mathbf{Z})$. By this we mean that $\mathrm{E}(X; \mathbf{Z}) = f(\mathbf{Z}_X) h_X(\mathbf{Z}')$ and $\mathrm{E}(Y; \mathbf{Z}) = g(\mathbf{Z}_Y) h_Y(\mathbf{Z}')$, where $\mathbf{Z}_X$ are the variables of $\mathbf{Z}$ that appear in $\mathrm{E}(X; \mathbf{Z})$ but not in $\mathrm{E}(Y; \mathbf{Z})$, and dually, $\mathbf{Z}_Y$ are those in $\mathrm{E}(Y; \mathbf{Z})$ that are not in $\mathrm{E}(X; \mathbf{Z})$. The variables $\mathbf{Z}'$ are those variables from $\mathbf{Z}$ not in $\mathbf{Z}_X \cup \mathbf{Z}_Y$. In these cases, the component of $\mathrm{Cov}(X, Y)$ that is contributed by the variation in $\mathbf{Z}$ (the extrinsic component) may be written as the covariance of the functions $h_X(\mathbf{Z}')$ and $h_Y(\mathbf{Z}')$. Conveniently, in the cases of interest here, the two functions $h_X$ and $h_Y$ coincide, and this is the form we use in the following decomposition principle. The supplementary material (Appendix A) contains the proof of this result.

### The noise decomposition principle (NDP)

Assume that there are measurable functions $f$, $g$, and $h$ such that $\mathrm{E}(X; \mathbf{Z})$ and $\mathrm{E}(Y; \mathbf{Z})$ split across common variables by way of $\mathrm{E}(X; \mathbf{Z}) = f(\mathbf{Z}_X) h(\mathbf{Z}')$ and $\mathrm{E}(Y; \mathbf{Z}) = g(\mathbf{Z}_Y) h(\mathbf{Z}')$. Then, provided that the variables $Z_1, \ldots, Z_m$ are mutually independent, the normalised covariance of $\mathrm{E}(X; \mathbf{Z})$ and $\mathrm{E}(Y; \mathbf{Z})$ will identify the total noise on $h(\mathbf{Z}')$ (i.e. $\eta_{h(\mathbf{Z}')}^2$).

As we show in the next section, there are many situations where the random variable $\mathrm{E}(X; \mathbf{Z})$ is precisely the common part of $\mathrm{E}(Y; \mathbf{Z})$ and $\mathrm{E}(X; \mathbf{Z})$ (i.e., $h(\mathbf{Z}') = \mathrm{E}(X; \mathbf{Z})$), and the normalised intrinsic contribution to the covariance is either zero or negligible. In these cases, the normalised covariance of $X$ and $Y$ will identify precisely the extrinsic noise contribution $\eta_{\mathrm{ext}}^2$ to the total noise $\eta_X^2$. To see this, consider the situation where $\mathrm{E}(Y; \mathbf{Z}) = f(\mathbf{Z}_Y) \mathrm{E}(X; \mathbf{Z})$. Then provided $f(\mathbf{Z}_Y)$ and $(X; \mathbf{Z})$ are independent random variables, the extrinsic contribution to the covariance of $X$ and $Y$ is given by,

$$\begin{aligned}
\mathrm{Cov}(\mathrm{E}(X;\mathbf{Z}),\mathrm{E}(Y;\mathbf{Z})) \; &= \mathrm{Cov}(\mathrm{E}(X;\mathbf{Z}),f(\mathbf{Z}_Y)\mathrm{E}(X;\mathbf{Z})) \\
&= \mathrm{E}(f(\mathbf{Z}_Y))\mathrm{Cov}(\mathrm{E}(X;\mathbf{Z}),\mathrm{E}(X;\mathbf{Z}_Y)) \\
&= \mathrm{E}(f(\mathbf{Z}_Y))\mathrm{Var}(\mathrm{E}(X;\mathbf{Z})).
\end{aligned} \tag{9}$$

If the normalised intrinsic contribution to the covariance is either zero or is negligible, it follows from (8) that

$$\frac{\mathrm{Cov}(X,Y)}{\mathrm{E}(X)\mathrm{E}(Y)} \approx \frac{\mathrm{Var}(\mathrm{E}(X;\mathbf{Z}))}{\mathrm{E}(X)^2} = \eta_{\mathrm{ext}}^2. \tag{10}$$

Thus, under certain conditions, measuring the covariance between two non-identically distributed and not-necessarily independent reporters can replace dual reporters.

## The pathway-reporter method

We show that for some reporters $X$ and $Y$ belonging to the same biochemical pathway, the covariance of $X$ and $Y$ continues to identify the extrinsic, and subsequently intrinsic, noise contributions to the total noise. The basis of the *pathway-reporter* method depends on the emergent covariances between the various species (e.g. nascent/mature mRNA and protein) in the gene expression pathway. Qualitatively, this effect can be seen in *Figure 3*, which compares simulated sample distributions of a simple four-stage model of gene transcription (refer to the model $\mathbf{M}_4$ below) in the case of moderate extrinsic noise to the case with no extrinsic noise. The plots are the bivariate distributions for nascent-mature, nascent-protein, and mature-protein levels, respectively. This will be made more

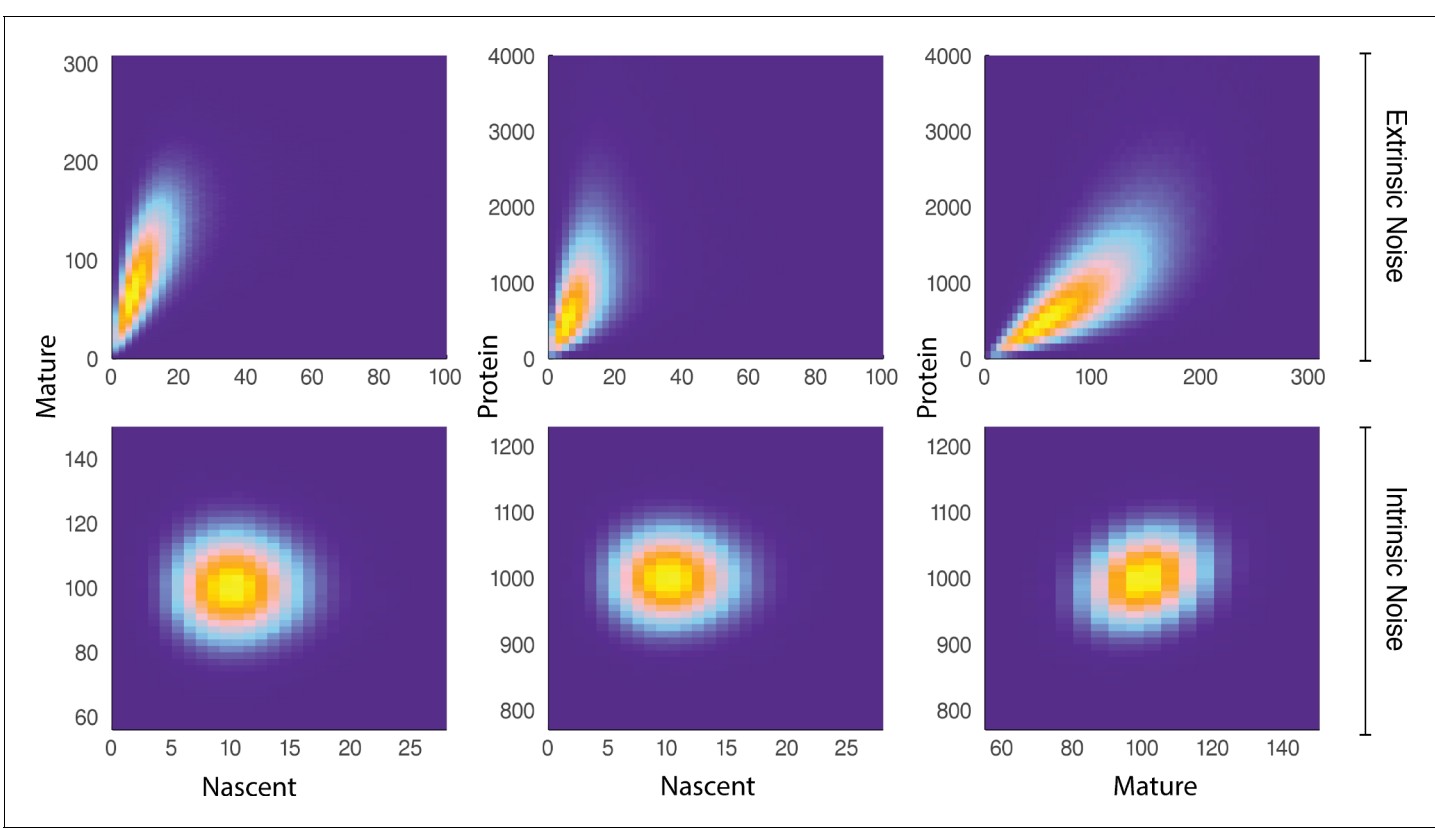

**Figure 3.** A comparison of joint distributions in the case of moderate extrinsic noise and no extrinsic noise. The plots are generated from a three-stage model of gene transcription, incorporating the production of nascent mRNA, mature mRNA and protein. Details of the model can be found in *Figure 4* (model $\mathbf{M}_4$) and the associated text. The top panel shows nascent-mature, nascent-protein and mature-protein joint distributions in the case of extrinsic noise, while the bottom panel displays the corresponding plots in the case of no extrinsic noise. Extrinsic noise produces a visibly more correlated joint distribution, which forms the basis of the pathway-reporter method.

precise below, where we find that it is possible to extract quantitative information about the extrinsic noise distribution itself from this data.

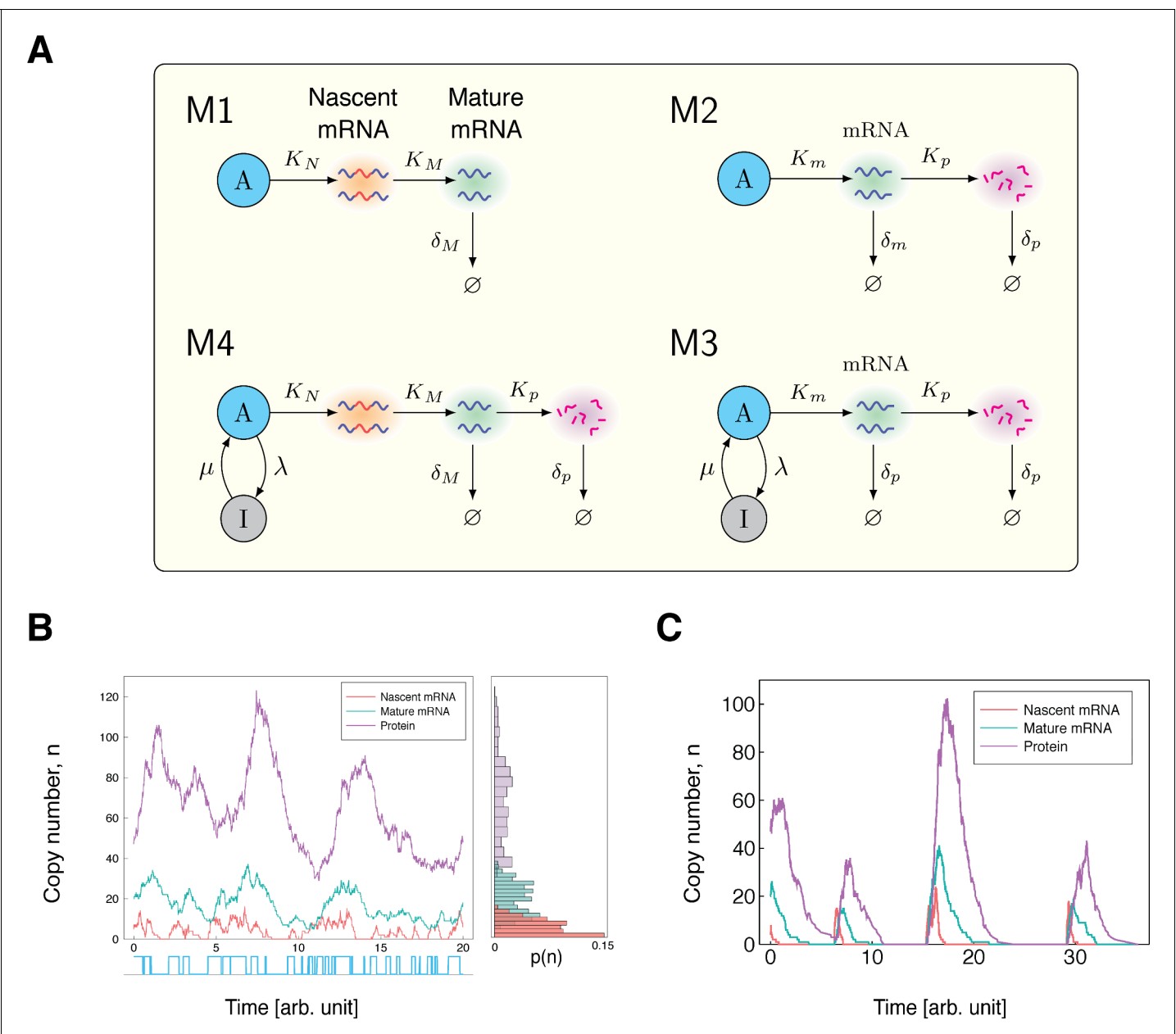

**Figure 4.** Stochastic models of gene expression. (A) The model $\mathbf{M}_1$ is the simplest model of mRNA maturation. Here, nascent (unspliced) mRNA are shown in red/blue wavy lines; the blue segments represent introns and the red segments represent the exons. Nascent mRNA are synthesised at the rate $K_N$, and spliced into mature mRNA (blue wavy lines) at rate $K_M$. Degradation of the mature mRNA occurs at rate $\delta_M$. The model $\mathbf{M}_2$ is the well-known two-stage model of gene expression. The model $\mathbf{M}_3$ is the extension of the two-stage model to include promoter switching. The nodes $A$ (active) and $I$ (inactive) represent the state of the gene, with transitions between states occurring at rates $\lambda$ and $\mu$. The remaining parameters are the same as those in the model $\mathbf{M}_2$. The model $\mathbf{M}_4$ extends the model $\mathbf{M}_3$ by incorporating mRNA maturation. Here, $K_N$ is the transcription rate parameter, and $K_M$ is the maturation rate. All other parameters are the same as in $\mathbf{M}_3$. (B) Time series simulation of the copy number and activity state of a gene modelled by $\mathbf{M}_4$. For ease of visualisation, the parameters were artificially chosen as $\lambda = 2$, $\mu = 2.5$, $K_N = 40$, $K_M = 4$, $K_p = 4$ and $\delta_p = 1$, with all parameters scaled relative to $\delta_m = 1$. (C) As $\lambda$ approaches 0, we see a higher correlation in the copy numbers of nascent mRNA, mature mRNA and protein. Again, the parameters are artificially chosen to be $\lambda = 0.1$, $\mu = 2.5$, $K_N = 80$, $K_M = 4$, $K_p = 4$ and $\delta_p = 1$, with all parameters scaled relative to $\delta_m = 1$.

Throughout this section, we assume that extrinsic noise sources act independently i.e., the environmental variables $Z_1, \ldots, Z_n$ of $\mathbf{Z}$ are mutually independent. Additionally, our modelling focuses only on a single gene copy, although the same analysis applies to multiple but indistinguishable gene copies; we refer to the supplementary material (Appendix B) for more details.

## Measuring noise from a constitutive gene

We consider first the simplest case where the underlying process is constitutive. We begin with a stochastic model of mRNA maturation, which we denote by $\mathbf{M}_1$; *Figure 4A* (top left) gives a schematic of the constitutive model. In this model, the gene continuously produces nascent mRNA according to a Poisson process at constant rate $K_N$, which are subsequently spliced into mature mRNA according at rate $K_M$. Degradation of mature mRNA occurs as a first-order Poisson process with rate $\delta_M$. The model $\mathbf{M}_1$, together with its extensions, has been considered in a number of recent studies *Gorin and Pachter, 2020a*; *Cao and Grima, 2020*; *La Manno et al., 2018*; *Gorin and Pachter, 2020b*; Bergen2020, and has a known solution for the stationary joint probability distribution *Jahnke and Huisinga, 2007* given by,

$$\tilde{p}(n,m;\theta) = \frac{e^{-\frac{K_n}{K_M}}\left(\frac{K_N}{K_M}\right)^n}{n!}\frac{e^{-\frac{K_n}{\delta}}\left(\frac{K_N}{\delta_M}\right)^m}{m!}, \tag{11}$$

where $n$ is the number of nascent mRNA, $m$ is the number of mature mRNA, and the parameter $\theta = (K_N, K_M, \delta_M)$. We use $X_N$ to denote the number of nascent mRNA, $X_M$ the number of mature mRNA produced from the same constitutive gene, and $\mathbf{Z} = (K_N, K_M, \delta_M)$. To simplify notation, we abbreviate the variables in $\mathbf{Z}_{X_N}$ as $\mathbf{Z}_N$, and similarly for $\mathbf{Z}_{X_M}$. It follows immediately from (11) that $X_N$ and $X_M$ are independent conditional on $\mathbf{Z}$, and so the intrinsic contribution to the covariance of $X_N$ and $X_M$ (the first term of (8)) is 0. It is also easy to see from (11) that $(X_N; \mathbf{Z}) = f(\mathbf{Z}_N)K_N$ and $(X_M; \mathbf{Z}) = g(\mathbf{Z}_M)K_N$, where $f(\mathbf{Z}_N) = \frac{1}{K_M}$ and $g(\mathbf{Z}_M) = \frac{1}{\delta_M}$. Since the extrinsic noise sources are assumed to act independently, it follows that the Noise Decomposition Principle (NDP) of the previous section holds. We then have that $\mathrm{Cov}(X_N, X_M) = \eta_{K_N}^2$, where $\eta_{K_N}^2$ is the total noise on the transcription rate parameter $K_N$. Thus, measuring $\mathrm{Cov}(X_N, X_M)$ can replace dual reporters to decompose the gene expression noise at the transcriptional level.

To support our mathematical results, we simulate the model $\mathbf{M}_1$ subject to parameter variation using the stochastic simulation algorithm (SSA). *Table 2* compares the extrinsic noise contributions found from various simulations with the corresponding theoretical values. In each simulation, the degradation rate $\delta_m$ is fixed at 1, with the other parameters scales accordingly. The maturation rate $K_M$ is sampled according to a $\mathrm{Gamma}(8, 0.0125)$ distribution, which has coefficient of variation 0.125. We consider different noise distributions on $K_N$, producing a range of noise strengths. Our theory predicts that pathway-reporters will identify the total noise on $K_N$. Overall, we observe an excellent agreement between the results obtained by the pathway-reporter method, the dual-reporter method and the theoretical noise. There is consistently slightly more variation in the pathway-reporter results compared with the dual-reporter results.

To explore the pathway-reporter method more widely, we consider 60 different parameter combinations to produce a range of mean copy numbers consistent with those reported experimentally. We also consider different noise distributions taken from the scaled Beta distribution family in order to produce a range of noise strengths; refer to *Supplementary file 1*. The pathway-reporter method performs favourably compared to the dual-reporter method calculated from mature mRNA, and consistently outperforms the dual-reporter method on nascent mRNA.

Next, we consider the simplest stochastic model of gene expression that includes both mRNA and protein dynamics: the well-known 'two-stage model' of gene expression, which, together with its three-stage extension to include promoter switching has been widely studied *Raser and O'Shea, 2005*; *Raj et al., 2006*; *Thattai and van Oudenaarden, 2001*; *Friedman et al., 2006*; *Singh and Hespanha, 2007*; *Shahrezaei and Swain, 2008a*; *Bokes et al., 2012*; *Molina et al., 2013*. We denote this model by $\mathbf{M}_2$; see *Figure 4A* (top right) for a schematic of this model. In this model, mRNA are synthesised according a Poisson process at rate $K_m$, which are then later translated into protein at rate $K_p$. Degradation of mRNA and protein occur as first-order Poisson processes with rates $\delta_m$ and $\delta_p$, respectively. If $X_m$ denotes the number of mRNA, $X_p$ the number of proteins

**Table 2.** A comparison of the pathway-reporter method and the dual-reporter method for constitutive expression under the model $\mathbf{M}_1$.

Here, PR (NM) gives the results of the nascent and mature pathway reporters, while DR (Mat) gives the results of dual reporters calculated from the mature mRNA. We considered noise on both the transcription rate ($K_N$) and the maturation rate ($K_M$). The decay rate is fixed at one, with the other parameters scaled accordingly. In each case, the maturation rate $K_M$ is varied according to a $\mathrm{Gamma}(8, 1.25)$ distribution, which has coefficient of variation 0.125. The values given are the average of 100 simulations, each calculated from 500 copy number samples, and the errors are ± one standard deviation. Our theory predicts that pathway-reporters will identify the noise on the nascent transcription rate $K_N$ ($\eta_{\mathrm{ext}}^2$). The noise distribution parameters are chosen to produce an average nascent mRNA copy number of approximately five and an average mature mRNA copy number of approximately 50.

| Theory | | Simulation | |
|---|---|---|---|
| (r)1-2 $\eta_{\mathrm{ext}}^2$ | Noise ($K_N$) | Pr (NM) | DR (Mat) |
| 0.00 | $K_N = 50$ | $0.00 \pm 0.01$ | $0.00 \pm 0.00$ |
| 0.10 | $\mathrm{Beta}_{133.\overline{3}}(6, 10.5)$ | $0.10 \pm 0.01$ | $0.10 \pm 0.01$ |
| 0.20 | $\mathrm{Gamma}(5, 10)$ | $0.20 \pm 0.02$ | $0.20 \pm 0.01$ |
| 0.50 | $\mathrm{Beta}_{300}(1.5, 7.5)$ | $0.49 \pm 0.04$ | $0.50 \pm 0.03$ |

The online version of this article includes the following source data for Table 2:

**Source data 1.** This is an Excel spreadsheet containing the data used to produce the final values in **Table 2**.

produced from the same constitutive gene, and if $\mathbf{Z} = (K_m, K_p, \delta_m, \delta_p)$, then the stationary means and covariance are given by *Thattai and van Oudenaarden, 2001*; *Singh and Hespanha, 2007*:

$$\mathrm{E}(X_m; \mathbf{Z}) = \frac{K_m}{\delta_m}, \quad \mathrm{E}(X_p; \mathbf{Z}) = \frac{K_p}{\delta_p}\frac{K_m}{\delta_m} \quad \text{and} \quad \mathrm{Cov}(X_m, X_p; \mathbf{Z}) = \frac{K_m K_p}{\delta_m(\delta_m + \delta_p)}. \tag{12}$$

It is easily verified that $\mathrm{E}(X_p; \mathbf{Z}) = f(\mathbf{Z}_p)\mathrm{E}(X_m; \mathbf{Z})$, where $f(\mathbf{Z}_p) = \frac{K_p}{\delta_p}$. Thus, it follows from the NDP that the normalised contribution of $\mathrm{Cov}(X_m, X_p)$ contributed by $\mathbf{Z}$ will identify the extrinsic noise contribution to the total noise on $X_m$. Now, if we assume that $\delta_m$ is fixed across the cell-population, and all parameters are scaled so that $\delta_m = 1$, we have the following expression for the intrinsic contribution to the covariance of $X_m$ and $X_p$; refer to the supplementary material (Appendix B) for details.

$$\frac{\mathrm{E}(\mathrm{Cov}(X_m, X_p; \mathbf{Z}))}{\mathrm{E}(X_m)\mathrm{E}(X_p)} = \frac{\alpha}{\mathrm{E}(K_m)}, \quad \text{where } \alpha = \frac{\mathrm{E}(1/(\delta_p + 1))}{\mathrm{E}(1/\delta_p)}. \tag{13}$$

Since mRNA tends to be less stable than protein, we have $\delta_p < 1$, and often $\delta_p \ll 1$ *Bernstein et al., 2002*; *Schwanhäusser et al., 2011*. So, we can expect $\alpha \ll 1$. Further, for many genes we can expect the number of mRNA per cell ($K_m$) to be in the order of tens, so $1/E(K_m) < 1$. It follows that $\mathrm{E}(\mathrm{Cov}(X_m, X_p; \mathbf{Z})) \ll \sim 1$, so that $\mathrm{Cov}(X_m, X_p)$ will closely approximate the extrinsic noise at the transcriptional level.

We test our theory using stochastic simulations of the model $\mathbf{M}_2$ subject to parameter variation. *Table 3* gives a comparison of the results of the mRNA-protein reporters and dual reporters. In each case, we varied $K_p$ according to a $\mathrm{Gamma}(5, 0.4)$ distribution and $\delta_p$ according to a $\mathrm{Gamma}(8, 0.125)$ distribution; the corresponding noise strengths are 0.20 and 0.125, respectively. We consider different noise distributions on $K_m$, which produce a range of noise strengths, and the noise distribution parameters are selected to produce a mean mRNA of approximately 50 and a mean number of approximately 1000 proteins in each simulation. As our theory predicts, the mRNA-protein reporters identify the extrinsic noise contribution to the total noise on $X_m$. Again, we can see from *Table 3* that there is excellent agreement between the results of the pathway reporters and the dual reporters, with slightly more variation in the pathway-reporter results. A larger exploration of the parameter space reveals similar results; these are provided in *Supplementary file 1*. Thus, despite mRNA and protein numbers not being strictly independent, they can, for practical purposes, replace dual reporters to decompose the noise at the transcriptional level.

**Table 3.** A comparison of the pathway-reporter method and the dual-reporter method for constitutive expression under the model $\mathbf{M}_2$.

Here PR (MP) gives the results of the mRNA-protien pathway reporters, while DR (Mat) gives the results of dual reporters calculated from the mature mRNA. We considered noise on the transcription rate ($K_m$), the protein synthesis rate ($K_p$), and the protein decay rate ($\delta_p$). The mRNA decay rate is fixed at one. In each case, we varied $K_p$ according to a $\mathrm{Gamma}(5, 0.4)$ distribution and $\delta_p$ according to a $\mathrm{Gamma}(8, 0.125)$ distribution; the corresponding noise strengths are 0.20 and 0.125, respectively. We considered different noise distributions on $K_m$, which produce a range of noise strengths. The noise distribution parameters are selected to produce a mean mRNA of approximately 50 and a mean number of approximately 1000 proteins in each simulation. The values given are the average of 100 simulations, each calculated from 500 copy number samples, and the errors are ± one standard deviation. As our theory predicts, the mRNA-protein reporters identify the noise on the transcription rate parameter $K_m$ ($\eta_{\mathrm{ext}}^2$).

| Theory | | Simulation | |
|---|---|---|---|
| (r)1-2 $\eta_{\mathrm{ext}}^2$ | Noise ($K_m$) | Pr (MP) | DR (Mat) |
| 0.00 | $K_m = 50$ | $0.00 \pm 0.01$ | $0.00 \pm 0.00$ |
| 0.10 | $\mathrm{Beta}_{133.\dot{3}}(6, 10.5)$ | $0.10 \pm 0.01$ | $0.10 \pm 0.01$ |
| 0.20 | $\mathrm{Gamma}(5, 10)$ | $0.20 \pm 0.02$ | $0.20 \pm 0.01$ |
| 0.50 | $\mathrm{Beta}_{300}(1.5, 7.5)$ | $0.51 \pm 0.04$ | $0.50 \pm 0.03$ |

The online version of this article includes the following source data for Table 3:

**Source data 1.** This is an Excel spreadsheet containing the data used to produce the final values in **Table 3**.

We note that both the pathway-reporter (nascent-mature or mature-protein) and dual-reporter methods show slower convergence when copy numbers are low. Pathway reporters usually show fractionally slower convergence and fractionally more variation than a dual reporter, as suggested by the standard deviations in **Tables 2** and **3**. A more detailed exploration of convergence is given in the supplementary material (Appendix C).

## Measuring noise from a facultative (bursty) gene

The most common mode of gene expression that is reported experimentally is burstiness *Jones and Elf, 2018*; *Raj et al., 2006*; *Bahar Halpern et al., 2015*; *Suter et al., 2011*; *Larsson et al., 2019*; *Golding et al., 2005*, in which mRNA are produced in short bursts with periods of inactivity in between. One example is gene regulation via repression, which naturally leads to periods of gene inactivity. Here, we consider a four-stage model of bursty gene expression, which incorporates both promoter switching and mRNA maturation; we denote this model by $\mathbf{M}_4$; see *Figure 4A* (bottom left). This model has recently been considered in *Cao and Grima, 2020*, where the marginal distributions are solved in some limiting cases. In this model, the gene switches probabilistically between an active state (A) and an inactive state (I), at rates λ (on-rate) and μ (off-rate), respectively. In the active state, nascent mRNA is synthesised according to a Poisson process at rate $K_N$, while in the inactive state transcription does not occur. Nascent mRNA is spliced into mature mRNA at rate $K_M$; this in turn is later translated into protein, with rate $K_P$. Degradation of mRNA and protein occur independently of the promoter state at rates $\delta_M$ and $\delta_P$, respectively.

For this model, we have three natural candidates for pathway reporters: (a) nascent and mature mRNA (b) mature mRNA and protein, and (c) nascent mRNA and protein reporters. Below we show that nascent mRNA–protein reporters yield consistently good estimates of the extrinsic noise contribution $\eta_{\mathrm{ext}}^2$ to the total gene expression noise, while nascent–mature and mature RNA–protein reporters are reliable in some restricted cases. We begin by showing that each of the reporter pairs (a), (b), and (c) satisfy the Noise Decomposition Principle. We then demonstrate computationally, that despite a lack of independence between these reporter pairs, the pathway-reporter method can still be used to decompose the total gene expression noise at the transcriptional level. Throughout, we let $X_N$ denote the number of nascent mRNA, we let $X_M$ denote the number of mature mRNA, and let $X_P$ denote the number of proteins produced from the same gene. We also let $\mathbf{Z} = \{\lambda, \mu, K_N, K_M, K_P, \delta_M, \delta_P\}$.

Following *Raj et al., 2006*, we assume that the transcription rate $K_N$ is large relative to the other parameters. We further assume that the maturation rate $K_M$ is large (i.e. $K_M > \delta_M$), which is supported by experiments *Cao and Grima, 2020*. Then, using the results of *Raj et al., 2006* and arguments similar to those in *Cao and Grima, 2020*, it can be shown that the stationary averages for the nascent mRNA, mature mRNA and protein levels are given by,

$$\mathrm{E}(X_N; \mathbf{Z}) = \frac{K_N}{K_M} \frac{\lambda}{(\lambda + \mu)}, \ \mathrm{E}(X_M; \mathbf{Z}) = \frac{K_N}{\delta_M} \frac{\lambda}{(\lambda + \mu)} \ \text{ and } \ \mathrm{E}(X_P; \mathbf{Z}) = \frac{K_P}{\delta_P} \frac{K_N}{\delta_M} \frac{\lambda}{(\lambda + \mu)}, \tag{14}$$

respectively. The supplementary material (Appendix B) provides more detail on how these expressions can be obtained.

We consider first the nascent-mature pathway reporters, case (a). From (14), it is easily seen that $\mathrm{E}(X_N; \mathbf{Z}) = f(\mathbf{Z}_N) K_N \frac{\lambda}{(\lambda + \mu)}$ and $\mathrm{E}(X_M; \mathbf{Z}) = g(\mathbf{Z}_M) K_N \frac{\lambda}{(\lambda + \mu)}$, where $f(\mathbf{Z}_N) = \frac{1}{K_M}$ and $g(\mathbf{Z}_M) = \frac{1}{\delta_M}$. So the NDP holds, and the normalised covariance of $\mathrm{E}(X_N; \mathbf{Z})$ and $\mathrm{E}(X_M; \mathbf{Z})$ will identify the extrinsic noise on the transcription component $K_N \frac{\lambda}{(\lambda + \mu)}$. Consider now the mature-protein reporters, case (b). Again, we can see from (14) that $\mathrm{E}(X_M; \mathbf{Z}) = f(\mathbf{Z}_P) \mathrm{E}(X; \mathbf{Z})$, where $f(\mathbf{Z}_P) = \frac{K_P}{\delta_P}$. Thus, the NDP holds, and so the normalised covariance of $\mathrm{E}(X_M; \mathbf{Z})$ and $\mathrm{E}(X_P; \mathbf{Z})$ will identify the total noise on $\mathrm{E}(X_M; \mathbf{Z})$ (the extrinsic noise on $X_M$). For the nascent-protein reporters, case (c), it is easy to see that $\mathrm{E}(X_N; \mathbf{Z}) = f(\mathbf{Z}_N) K_N \frac{\lambda}{(\lambda + \mu)}$, where $f(\mathbf{Z}_N) = \frac{1}{K_M}$, and $\mathrm{E}(X_P; \mathbf{Z}) = g(\mathbf{Z}_P) K_N \frac{\lambda}{(\lambda + \mu)}$, where $g(\mathbf{Z}_P) = \frac{K_P}{\delta_M \delta_P}$. Thus, again the NDP holds, and the normalised covariance of $\mathrm{E}(X_N; \mathbf{Z})$ and $\mathrm{E}(X_P; \mathbf{Z})$ will identify the noise on the transcriptional component $K_N \frac{\lambda}{(\lambda + \mu)}$.

In order for the pathway-reporter method to provide a close approximation to the extrinsic noise in cases (a), (b), and (c), we require that the normalised intrinsic contribution to the covariance is either zero or negligible. This condition will hold provided there is sufficiently small correlation between the reporter pairs. In the case of (prokaryotic) mRNA and protein, this lack of correlation has been been verified experimentally in *E. coli* (*Taniguchi et al., 2010*). More generally, it is possible to provide an intuition about the conditions under which the lack of correlation might hold. The time series of copy numbers for each of nascent mRNA, mature mRNA and protein broadly follow each other, each with delay from its predecessor (*Figure 4B*). Parameter values that reduce this delay will tend to increase correlation, and thereby increase the normalised intrinsic contribution to the covariance. The primary example of this effect is seen when $\delta_p$ approaches, or even exceeds $\delta_m$ (or for nascent-mature reporters, when $\delta_M$ approaches the maturation rate). A further contributor to high correlation between mRNA and protein, is when the system undergoes long timescale changes. In this situation, the copy numbers tend to drop to very low values for extended periods. The primary parameter influencing this type of behaviour is the active-rate $\lambda$, specifically, when $\lambda$ tends to 0 (*Figure 4C*). An illustrative example of this can be seen by considering a Telegraph system in the limit of slow switching, which produces a copy number distribution that converges to a scaled Poisson-Bernoulli compound distribution: even without any extrinsic noise, the pathway reporter method will identify the $\eta^2$ value of the corresponding scaled Bernoulli distribution.

An extensive computational exploration of the parameter space (*Supplementary file 2*) supports our intuition, though the strength of the effect varies across the three different reporter pairs. This is further corroborated by the heatmaps shown in *Figure 5*, which for three fixed values of $\mu$ and a broad spectrum of $\delta_p$ and $\lambda$ values, give the intrinsic term $\eta^2_{\mathrm{int}}$ in (8), for fixed $\mathbf{Z}$. Thus, the heatmaps provide an estimate for the overshoot error in the pathway-reporter approach. Note that blue pixels represent an overshoot estimate of less than 0.05.

For nascent-protein reporters, the normalised intrinsic contribution to the covariance is satisfactorily small (less than 0.05) except for (a) high values of $\delta_p$ in unison with (b) low values of $\lambda$ (less than 1, although lower values are acceptable if $\delta_p$ is small). The assumption (a) that $\delta_P < \delta_M$ is known to be true for a large number of genes, and is justified by the difference in the mRNA and protein lifetimes. While there is of course variation across genes and organisms, values of $\delta_P \leq 0.5 \delta_M$ and even $\delta_P \leq 0.2 \delta_M$ seem reasonable for the majority of genes. In *E. coli Taniguchi et al., 2010* and yeast *Belle et al., 2006*, for example, mRNA are typically degraded within a few minutes, while most proteins have lifetimes at the level of 10 s of minutes to hours. For mammalian genes *Schwanhäusser et al., 2011*, it has been reported that the median mRNA decay rate $\delta_M$ is (approximately) five times larger than the median protein decay rate $\delta_P$, determined from 4200 genes.

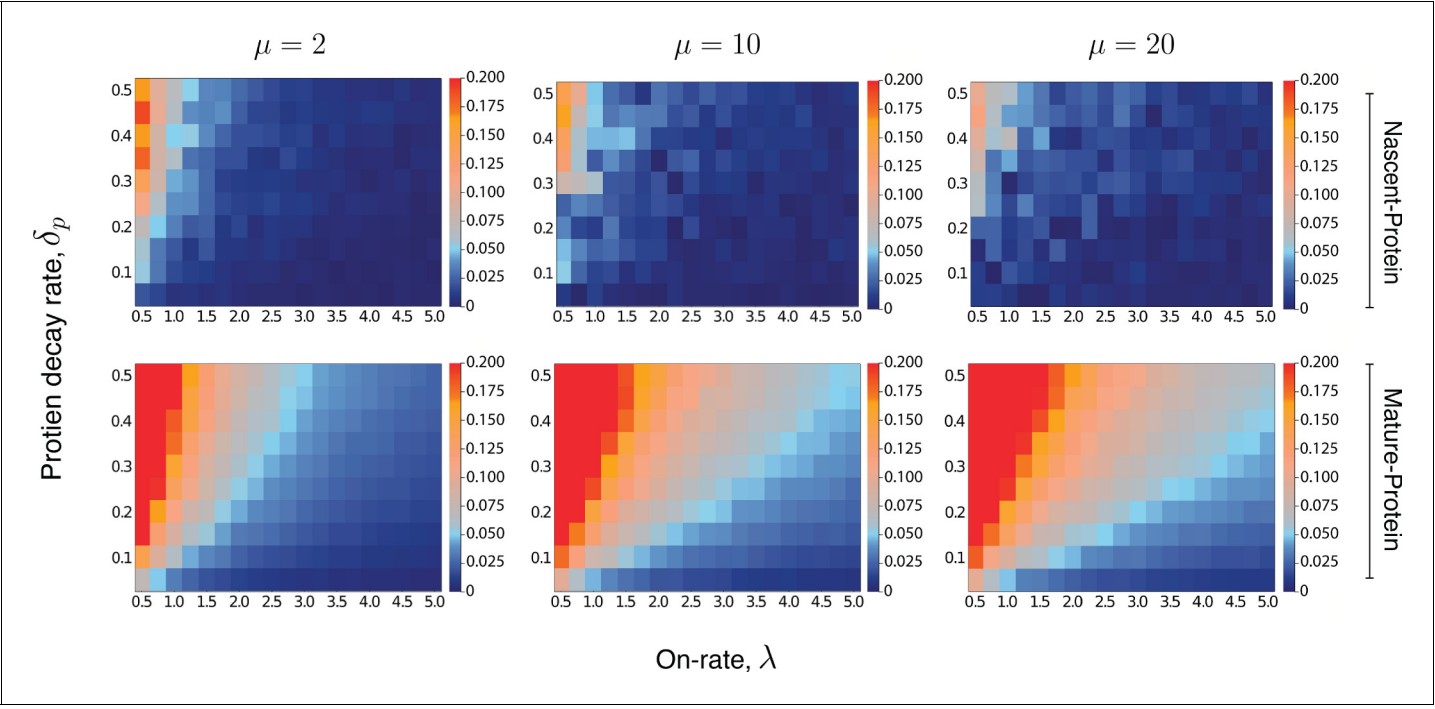

**Figure 5.** Heatmaps for the intrinsic contribution to the covariance. These heatmaps estimate the level of overshoot in the pathway-reporter approach for the nascent-protein and mature-protein reporters; blue regions show an overshoot of less than $\simeq 0.05$. Here, the intrinsic contribution is calculated using stochastic simulations of the model $\mathbf{M}_4$. For the mature-protein and nascent-protein reporters, we consider three different values of the parameter μ, specifically $\mu = 2$, $\mu = 10$, and $\mu = 20$. In all cases, the parameter $\delta_p$ and the on-rate $\lambda$ are varied between 0.01 and 0.5, and 0.5 and 5, respectively. The parameters of the model $\mathbf{M}_4$ are scaled so that $\delta_M = 1$. The maturation rate is fixed at 20, with the parameters $K_N$ and $K_P$ chosen to produce a mean protein level of 1000, a mean nascent mRNA level of 5 and a mean mature mRNA level of 50. Each individual pixel is generated from a sample of size 3000, although there is still some instability in the convergence for the nascent-protein reporter, particularly as the overshoot estimation starts to increase, and particularly as μ is larger. To produce more accurate values, the case of $\mu = 2$ was averaged over two full experiments while $\mu = 20$ was averaged over three. This was also done for the mature-protein reporter, however for these images there was almost no visible difference between the various runs of the experiment and their averages. Each of the three μ values takes approximately 7–10 hr of computation, depending on lead in time before sampling within a simulation. *Figure 5—figure supplement 1* gives a heatmap for the overshoot in the pathway-reporter approach for nascent-mature pathway reporters.

The online version of this article includes the following figure supplement(s) for figure 5:

**Figure supplement 1.** Heatmap for the intrinsic contribution to the covariance for nascent-mature pathway reporters.

Assumption (b) requires that the gene is sufficiently active. In a recent paper by *Larsson et al., 2019*, the promoter-switching rates $\lambda$, μ, and the transcription rate $K$ of the Telegraph model are estimated from single-cell data for over 7000 genes in mouse and human fibroblasts. Of those genes with mean mRNA levels greater than 5, we found that over 90% have a value for $\lambda$ of at least 0.5, and over 65% have a value for $\lambda$ greater than 1. In *Cao and Grima, 2020*, a comprehensive list of genes (ranging from yeast cells through to human cells) with experimentally inferred parameter values are sourced from across the literature (see *Table 1* in *Cao and Grima, 2020*). After scaling the parameter values of the 26 genes reported there, we find that around 88% have a value for $\lambda$ of at least 0.5, and approximately 58% have a value for $\lambda$ greater than 1. Thus, the heatmaps given in *Figure 5* (top panel) suggest that nascent-protein reporters will provide a satisfactory estimate of the extrinsic noise level for a substantial fraction of genes.

The mature nRNA–protein reporters are less reliable, with the requirement of slow protein decay and higher on-rate being more pronounced than for the nascent mRNA–protein reporters; this is evident from *Figure 5* (second panel). The performance of the nascent–mature reporter is of course independent of $\delta_p$, but is only viable in the case of a large on-rate (see *Figure 5—figure supplement 1*).

We test our approach for each of the pathway reporter pairs (a), (b), and (c) against dual reporters using stochastic simulations. *Table 4* shows the results from six simulations across a spectrum of behaviours from moderately slow switching, to fast switching as well as a range of levels of burstiness. For each of the parameters $\mu, \lambda, K_P, \delta_P$, we selected a scaled $\mathrm{Beta}(5, 6)$ distribution, with squared coefficient of variation $\eta^2 = 0.1$; the scaling is chosen in each case to achieve a mean value equal to the parameter value given in *Table 4*. It is routinely verified that scaling these distributions does not change the value of $\eta^2$. The parameter $K_N$ is given the noise distribution $\mathrm{Beta}(3, 6)$, which has a slightly higher coefficient of variation $\eta^2 = 0.2$. In order to achieve direct benchmarking against the dual reporters, the parameter $K_M$ is fixed; we select $K_M = 10$. This is because the nascent-protein pathway reporter estimates noise on the value of $K_N \frac{\lambda}{\lambda + \mu}$, while the mature mRNA dual-reporter measures noise on $\frac{K_N}{K_M} \frac{\lambda}{\lambda + \mu}$, and these coincide only when $K_M$ is fixed. The mean values of $K_N$ and $K_P$ are chosen to achieve approximate average nascent mRNA levels, mature mRNA levels and protein levels at 5, 50, and 1000 respectively, given the chosen values of $\lambda, \mu, \delta_P$.

The results for the nascent mRNA–protein reporters, case (c), given in *Table 4* show comparable performance to dual reporters, with only modest overshoot; even in the worst performing case of $\lambda = 0.5$, $\mu = 1$ the result of the pathway reporters is within one standard deviation, in a very tight distribution. The error heatmaps of *Figure 5* provide an accurate estimate of the overshoot in the nascent-protein results in *Table 4*. As an example, the first row is most closely matched by the heatmap at top left of *Figure 5*, which at $\lambda = 0.5$ and $\delta_P = 0.1$ is suggestive of an error around the boundary between blue and red (around 0.06). The same accuracy is obtained for the other rows. As predicted, the mature-protein reporters show significantly more overshoot, especially with the less active genes. Improved accuracy can again be obtained by subtracting the estimated overshoot given in the error heatmaps from the obtained value. Thus for example, the error heatmap for $\mu = 2$ (*Figure 5* lower left) gives an error approximately 0.07 for $\lambda = 1, \delta_P = 0.1$, which agrees very closely to the actual overshoot of 0.07 shown in the corresponding row of *Table 4*. An overshoot of approximately 0.06 is suggested by the heatmap for $\mu = 2$, when $\lambda = 2, \delta_P = 0.3$, which leads to a correction from 0.35 in *Table 4* to a value of 0.29. This is quite consistent with the dual reporter benchmark of 0.27. As expected (based on *Figure 5—figure supplement 1*), nascent-mature reporters do not perform well on bursty systems except for high $\lambda$ and so the values are not included in *Table 4*; only in

**Table 4.** A comparison of the pathway-reporter method and dual-reporter method for bursty expression.

Here PR (NP) gives the results of the nascent and protein pathway reporters, PR (MP) gives the results of the mRNA and protein reporters, while DR (Mat) gives the results of the dual reporters calculated from the mature mRNA. We consider noise on all of the parameters except for $\delta_M$ and $K_M$; see discussion in main text. The values given are the average of 100 simulations, each calculated from 500 copy number samples, and the errors are $\pm$ one standard deviation. Our theory predicts that pathway-reporters will identify the noise at both the promoter level ($\lambda, \mu$) and transcriptional level ($K_N$); the total extrinsic noise in each case is given by $\eta_{\mathrm{ext}}^2$. As before, the noise distribution parameters are chosen to produce an average nascent mRNA copy number of 5 and an average mature mRNA copy number of 50, and an average number of 1000 proteins.

| Mean | | | | | Simulation | | |
|---|---|---|---|---|---|---|---|
| (r)1-5 $\lambda$ | $\mu$ | $K_N$ | $K_P$ | $\delta_P$ | Pr (MP) | Pr (NP) | DR (Mat) |
| 0.5 | 1 | 150 | 2 | 0.1 | $0.46 \pm 0.06$ | $0.38 \pm 0.07$ | $0.32 \pm 0.07$ |
| 1 | 2 | 150 | 2 | 0.1 | $0.39 \pm 0.05$ | $0.34 \pm 0.07$ | $0.32 \pm 0.05$ |
| 1 | 20 | 1050 | 2 | 0.1 | $0.66 \pm 0.15$ | $0.52 \pm 0.22$ | $0.50 \pm 0.15$ |
| 2 | 2 | 100 | 6 | 0.3 | $0.35 \pm 0.04$ | $0.29 \pm 0.05$ | $0.27 \pm 0.03$ |
| 2 | 20 | 550 | 6 | 0.3 | $0.61 \pm 0.09$ | $0.47 \pm 0.15$ | $0.47 \pm 0.09$ |
| 10 | 10 | 100 | 6 | 0.3 | $0.29 \pm 0.03$ | $0.27 \pm 0.04$ | $0.27 \pm 0.02$ |

The online version of this article includes the following source data for Table 4:

**Source data 1.** This is an Excel spreadsheet containing the data used to produce the final values in *Table 4*.

the case of $\lambda = \mu = 10$ does the result begin to approach the dual reporter value, returning $0.32 \pm 0.03$.

## Generality of the pathway reporter method

To test the robustness of our pathway reporter approach, we validate our theoretical results on various other gene expression dynamics. (1): We begin by considering a more detailed model of the mRNA maturation process, where the nascent mRNA maturate after a fixed amount of time. The assumption of a fixed maturation time is sometimes taken to approximate the combined effect of intermediate maturation processes such as initiation, elongation, and release *Xu et al., 2016*. More specifically, we consider the model $\mathbf{M}_4$ above (*Figure 4D*) in the case of constitutive expression ($\lambda = 1$, $\mu = 0$), and replacing the first-order reaction $K_M$ by a fixed interval of time. Additionally, we explore maturation times sampled from Erlang distributions, to account for the fact that maturation can involve several shorter stochastic processes. We find that the extrinsic noise contribution obtained using the nascent and mature mRNA reporters match closely to the true (dual reporter) values across a range of maturation times; refer to the supplementary material for details.

(2): Next we consider an extension of a model of transcriptional bursting given in *Bartman et al., 2019*; *Cao et al., 2020*. The model we consider is the same as in *Bartman et al., 2019*; *Cao et al., 2020*, however, is extended to include protein synthesis and degradation. This model captures the transcriptional process at a finer level of detail, and is argued in *Cao et al., 2020* to be the model most closely matching experimental observations. In this more nuanced 'multiscale' model, the gene stochastically switches between three states: two active states $S_{10}$ and $S_{11}$, and one inactive state $S_0$. The activation of the gene occurs in two steps, initially by the binding of transcriptional factors (transition from $S_0$ to $S_{10}$ at rate $\lambda_1$, and reversible at rate $\mu_1$), and then as a secondary step, by the binding and pause of the mRNA polymerase (transition from $S_{10}$ to $S_{11}$ at rate $\lambda_2$). Transitions from $S_{11}$ to $S_0$ also occur at rate $\mu_1$, due to detachment of both the transcriptional factors and polymerase. Transcription of nascent mRNA (at rate $K_N$) occurs only in state $S_{11}$ and results in immediate transition to state $S_{10}$. Nascent mRNA maturate at rate $K_M$, and are subsequently translated into protein at rate $K_p$. Degradation of mRNA and protein occur with rates $\delta_m$ and $\delta_p$, respectively. All reactions are considered to be first-order with exponentially distributed waiting times between successive reactions. A schematic of the model is given in *Figure 6* (inner rectangle). In this case, we are again able to prove that the Noise Decomposition Principle holds for all reporter pairs taken from this pathway using existing formulæ derived in *Cao et al., 2020* for the marginal means. For details refer to the supplementary material (Appendix Convergence of Pathway and Dual Reporters).

(3): We combine models (1) and (2) above, incorporating the fixed time maturation of (1) with the multiscale approach of (2).

(4): The cell cycle is a major contributing factor to extrinsic noise, introducing population heterogeneity (as cells are at different stages of the cell cycle), as well as internal dynamics to parameter values. Here we incorporate the salient features of the cell cycle into model (3), which is measurable as extrinsic noise by our methodology. Specifically, we model the effects of (i) gene replication, (ii) dosage compensation, (iii) binomial partitioning of products due to cell division, as well as (iv) cell-cycle length variability. Refer to *Figure 6* for a schematic. This model is an extension of that solved in *Cao and Grima, 2020*. Our model further incorporates the multi-scale dynamics of model (3) and the Erlang-distributed maturation times of model (1). As far as we are aware, this model has not been explored even by stochastic simulations before. A detailed description of how the above cell-cycle effects are captured in our model is given as follows. (i) Replication results in a doubling of the gene from one to two at the replication time, $t_r$. This replication occurs over a period which is much shorter than the length of the cell cycle, and we follow the assumptions in *Cao and Grima, 2020* by considering it to occur instantaneously. (ii) Dosage compensation changes the rate at which the gene switches from inactive to active ($\lambda_1$) upon replication at time $t_r$. Again following *Cao and Grima, 2020*, the activation rate after replication is 70% of the rate prior to replication. (iii) Binomial partitioning of nascent mRNA, mature mRNA and protein at cell division. We assume that nascent mRNA, mature mRNA, and protein segregate into the two daughter cells, with independent probability $1/2$. We follow just one of the daughter cells, chosen at random with equal probability. (iv) Cell-cycle length variability. The time between successive cell divisions is stochastic, and is assumed to be sampled from an Erlang distribution. This has been shown to be consistent with experimental data *Cao and Grima, 2020*. The time to replication, and subsequently to cell division, are both

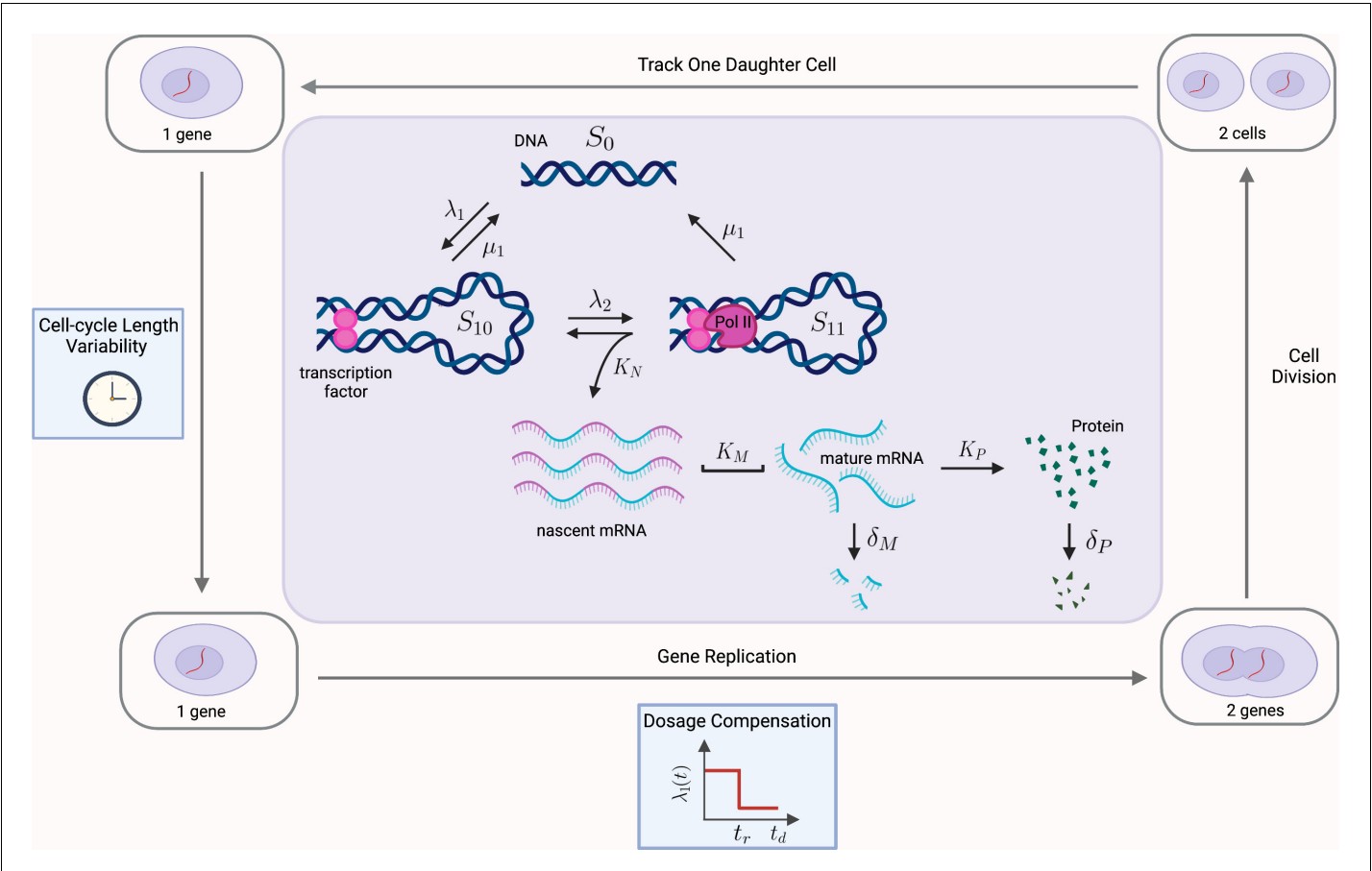

**Figure 6.** Multiscale model of transcriptional bursting with additional features of the cell cycle. In this model, the gene stochastically switches between three states: two active states, $S_{10}$ and $S_{11}$, and one inactive state $S_0$. Gene activation occurs in two steps, initially by the binding of transcription factors (at rate $\lambda_1$, reversible at rate $\mu_1$), and then as a secondary step by the binding and pause of the mRNA polymerase (at rate $\lambda_2$). Transitions from $S_{11}$ to $S_0$ also occur at rate $\mu_1$, due to detachment of both the transcriptional factors and polymerase. Transcription of nascent mRNA (at rate $K_N$) occurs only in state $S_{11}$ and results in immediate transition to state $S_{10}$. Nascent mRNA mature at rate $K_M$, and are subsequently translated into protein at rate $K_p$. Degradation of mRNA and protein occur with rates $\delta_m$ and $\delta_p$, respectively. We verify our pathway reporter method on three variations of the multiscale model. First, we assume all reactions are first-order Poisson processes (Case (2) in the main text). We then incorporate further details of the mRNA maturation process, where nascent mRNA occurs after a fixed amount of time (Case (3)). Finally, we incorporate features of the cell-cycle such as gene replication, dosage compensation, cell division, and cell-cycle length variability, as well as incorporating more realistic Erlang distributed maturation times (Case (4)).

chosen from an Erlang distribution with shape parameter 12, which produces a total cell cycle length distributed according to $\mathrm{Erlang}(24, \lambda)$; this matches the $\mathrm{Erlang}(24, \lambda)$ cell cycle length in **Cao and Grima, 2020**, where replication is at the exact halfway point. We similarly choose a rate parameter $\lambda = \lambda_1$ chosen at a commensurate proportion to our mRNA decay rate of $\delta = 1$.

In each of the cases (1 – 4) above, we find that the correlations between reporter pairs is negligible, and the predicted contribution of extrinsic noise matches those obtained from the dual reporter method across a range of parameter combinations. Details of the simulation methods and results can be found in the supplementary material (Appendix Convergence of Pathway and Dual Reporters). In summary, the results show that our pathway reporter approach is remarkably insensitive to the specific dynamics of mRNA and protein synthesis. In particular, the correlations between reporter pairs do not strongly depend on the details of the gene expression model used.

## Discussion

Despite the proliferation of experimental methods for single-cell profiling, the ability to extract transcriptional dynamics from measured distributions of mRNA copy numbers is limited. In particular, the multiple factors that contribute to mRNA heterogeneity can confound the measured distribution, which hinders analysis. Theoretical innovations that allow us to quantify and help in identifying the causes of these observable effects are therefore of great importance. In this work, we have demonstrated, through a series of mathematical results, that it is impossible to delineate the relative sources of heterogeneity from the measured transcript abundance distribution alone: multiple possible dynamics can give rise to the same distribution. Our approach involves establishing integral representations for distributions that are commonly encountered in single-cell data analysis, such as the negative binomial distribution and the stationary probability distribution of the Telegraph model. We show that a number of well-known representations can be obtained from our results. A particular feature of our non-identifiability results is that population heterogeneity inflates the apparent burstiness of the system. It is therefore necessary to collect further information, beyond measurements of the transcripts alone, in order to constrain the number of possible theoretical models of gene activity that could represent the system. In particular, additional work may be required to determine the true level of burstiness of the underlying system.

We have developed a theoretical framework for estimating levels of extrinsic noise, which can assist in resolving non-identifiability problems. The dual reporter method of *Swain et al., 2002* already provides one such approach; but it is experimentally challenging to set up in many systems, and requires strictly identical and conditionally independent pairs of gene reporters. Our *Noise Decomposition Principle* directly generalises the theoretical underpinnings of the dual reporter method and related approaches *Bowsher and Swain, 2012*; *Jetka et al., 2018*; here we have used it to identify a practical approach—the pathway-reporter method—for obtaining an effective and experimentally more easily implementable noise decomposition. Our approach allows us to use measurements of two different species from the transcriptional pathway of a single gene copy instead of having to set up a more cumbersome dual reporter assay. The accuracy of the pathway-reporter method is provably identical for constitutive gene expression, and in the case of nascent-mature mRNA reporters, the measurements are readily obtainable from current single-cell data *Shah et al., 2018*; *La Manno et al., 2018*; *Skinner et al., 2016*. For bursty systems, the method in general provides only an approximation of the extrinsic noise. We are, however, able to demonstrate computationally, that one of the proposed pathway reporters provides a satisfactory estimate of the extrinsic noise for most genes. The other pathway reporters also provide viable estimates of the extrinsic noise in some cases. We further validate our theoretical framework on synthetic data for genes with various underlying gene expression dynamics. Our results show that the pathway reporter method is independent of the specific dynamics of mRNA and protein synthesis, and therefore should be applicable to a broad range of experimental scenarios.

Despite the generality of our theoretical contribution, our pathway-reporter approach has some caveats. In particular, the approach relies on the assumption that extrinsic noise sources act independently. Experimentally, however, these may be correlated. For example, it has been suggested *Hilfinger et al., 2016*; *Cole and Luthey-Schulten, 2017* that the transcription and translation rates in *E. coli* anticorrelate. Additional work is required to determine degree to which the independence of noise sources is a reasonable assumption.

Recent developments in single-cell profiling now allow simultaneous measurements of transcripts and proteins in thousands of single cells *Stoeckius et al., 2017*; *Peterson et al., 2017*; *Reimegård et al., 2019*. As discussed in *Gorin and Pachter, 2020a*, experimental improvements that would additionally allow measurements of nascent transcripts, coupled with theoretical developments such as those presented here, will allow for identification of noise sources on a genome-wide scale. Our work reveals that extrinsic noise distorts the multivariate copy number distribution of the different species in the gene expression pathway. We have exploited this to yield reliable estimates of noise strength, which we are confident will assist in setting better practices for model fitting and inference in the analysis of single-cell data. A more nuanced analysis of this multivariate distribution may offer even further insight into model and noise identifiability.

## Acknowledgements

The authors gratefully acknowledge Rowan D Brackston for helpful discussions in the early stages of this research. We thank Lior Pachter and Gennady Gorin for fruitful discussions on noise identifiability. We also wish to thank Arjun Raj for providing valuable feedback on this work. LH. and MPHS. were supported by the University of Melbourne Driving Research Momentum initiative.

## Additional information

### Funding

| Funder | Grant reference number | Author |
|---|---|---|
| University of Melbourne | DRM | Lucy Ham<br>Michael PH Stumpf |
| Volkswagen Foundation | 93 062 | Michael PH Stumpf |

The funders had no role in study design, data collection and interpretation, or the decision to submit the work for publication.

### Author contributions

Lucy Ham, Conceptualization, Resources, Software, Formal analysis, Validation, Investigation, Visualization, Methodology, Writing - original draft, Writing - review and editing; Marcel Jackson, Conceptualization, Software, Formal analysis, Validation, Investigation, Methodology, Writing - original draft, Writing - review and editing; Michael PH Stumpf, Conceptualization, Supervision, Funding acquisition, Methodology, Writing - original draft, Project administration, Writing - review and editing

### Author ORCIDs

Lucy Ham  https://orcid.org/0000-0001-5177-4434
Marcel Jackson  https://orcid.org/0000-0002-8149-1141
Michael PH Stumpf  https://orcid.org/0000-0002-3577-1222

### Decision letter and Author response

Decision letter https://doi.org/10.7554/eLife.69324.sa1
Author response https://doi.org/10.7554/eLife.69324.sa2

## Additional files

### Supplementary files

• Supplementary file 1. Simulation results of the pathway-reporter method for constitutive genes across 60 different parameter values. We consider noise on all of the parameters except mRNA decay in a constitutive model with mRNA maturation and protein translation. Refer to the excel spreadsheet ConstitutiveaResults.xlsx for full details of the simulation, including the chosen noise distributions and parameters.

• Supplementary file 2. Simulation results for the overshoot estimate in the pathway-reporter method for bursty genes across 448 different parameter values. Refer to the excel spreadsheet NoiseFreeaResults.xlsx for full details of the simulation, including the chosen noise distributions and parameters.

• Transparent reporting form

### Data availability

All methods and simulation results are shared via a github site https://github.com/leham/PathwayReporters (copy archived at https://archive.softwareheritage.org/swh:1:rev:269e0fffe4fc716db6991cc-f78ad2191e509c2e1). There is no original data associated with this manuscript.

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

## Appendix 1

### Derivations of the non-identifiability results

In the main text we provide a number of examples of when the compound distribution does not have a unique representation. We here provide the full details of the derivations of these results.

### Bursty expression: telegraph representation

Consider first a Telegraph distribution $\tilde{p}_T(n;\lambda,\mu',K')$ and the probability density function of the scaled beta distribution $\text{Beta}_{K'}(\lambda+\mu,\mu'-\mu)$, given by

$$f_{K'}(t;\lambda+\mu,\mu'-\mu) = \frac{\Gamma(\lambda+\mu')}{\Gamma(\lambda+\mu)\Gamma(\mu'-\mu)} K'^{1-\lambda-\mu'} t^{\lambda+\mu-1} (K'-t)^{\mu'-\mu-1}.$$

Note that this distribution has support $[0,K']$. In the main text, we claim that the Telegraph distribution $\tilde{p}_T(n;\lambda,\mu',K')$ can be obtained from compounding a Telegraph distribution by a scaled beta distribution with pdf $f(t;\lambda+\mu,\mu'-\mu)$. In other words, that $\tilde{p}_T(n;\lambda,\mu',K')$ can be written as:

$$\tilde{p}_T(n;\lambda,\mu',K') = \int_0^{K'} \tilde{p}_T(n;\lambda,\mu,t) f_{K'}(t;\lambda+\mu,\mu'-\mu)\, dt. \tag{15}$$

This is *Equation (3)* in the main text. Starting from the right hand side of (15), we have

$$\int_0^{K'} \tilde{p}_T(n;\lambda,\mu,t) f_{K'}(t;\lambda+\mu,\mu'-\mu)\, dt$$
$$= \frac{1}{n!} \frac{\Gamma(\lambda+\mu')}{\Gamma(\lambda+\mu)\Gamma(\mu'-\mu)} \frac{\Gamma(\lambda+n)}{\Gamma(\lambda)} \frac{\Gamma(\lambda+\mu)}{\Gamma(\lambda+\mu+n)}$$
$$\int_0^{K'} {}_1F_1(\lambda+n,\lambda+\mu+n,-t)(K')^{1-\lambda-\mu'} t^{\lambda+\mu+n-1} (K'-t)^{\mu'-\mu-1}\, dt. \tag{16}$$

Substituting $t = K'T$ and simplifying, the right hand side of (16) becomes

$$\frac{1}{n!} \frac{\Gamma(\lambda+\mu')}{\Gamma(\mu'-\mu)} \frac{\Gamma(\lambda+n)}{\Gamma(\lambda)} \frac{(K')^n}{\Gamma(\lambda+\mu+n)}$$
$$\int_0^1 {}_1F_1(\lambda+n,\lambda+\mu+n,-KT)T^{\lambda+\mu+n-1}(1-T)^{\mu'-\mu-1}\, dT.$$

Now, using the integral representation *Olver et al., 2010* [13.4.2] of the confluent hypergeometric function (with $a=\lambda+n$, $b=\lambda+\mu'+1$ and $c=\lambda+\mu+1$), this becomes

$$\frac{1}{n!} \frac{\Gamma(\lambda+\mu')}{\Gamma(\mu'-\mu)} \frac{\Gamma(\lambda+n)}{\Gamma(\lambda)} \frac{(K')^n}{\Gamma(\lambda+\mu+n)} \frac{\Gamma(\lambda+\mu+n)\Gamma(\mu'-\mu)}{\Gamma(\mu'+\lambda+n)} {}_1F_1(\lambda+n,\lambda+\mu'+n,-K')$$
$$= \frac{\Gamma(\lambda+\mu')}{\Gamma(\lambda+\mu'+n)} \frac{\Gamma(\lambda+n)}{\Gamma(\lambda)} \frac{(K')^n}{n!} {}_1F_1(\lambda+n,\lambda+\mu'+n,-K') = \tilde{p}_T(n;\lambda,\mu',K'),$$

which is the left hand side of (15). Hence, we have that (15) holds.

### Instantaneously bursty expression: negative binomial representations

We consider first the representation given in *Equation (4)* in the main text. Recall that we let $\tilde{p}_{\text{NB}}(n;r,\beta)$ denote the probability mass function of a $\text{NegBin}(r,\frac{\beta}{\beta+1})$ distribution, where $\beta > \sim 0$. In the main text, we claim that for any negative binomial distribution of the form $\text{NegBin}(\lambda,\frac{\beta}{\beta+1})$ (where $\beta > 0$) we have,

$$\tilde{p}_{\text{NB}}(n;\lambda,\beta) = \int_0^\infty \tilde{p}_T(n;\lambda,\mu,t) f(t;\lambda+\mu,\beta)\, dt, \tag{17}$$

where $f(t;\lambda+\mu,\beta)$ is the probability mass function of a $\text{Gamma}(\lambda+\mu,\beta)$ distribution. Beginning with the right hand side of (17) we have,

$$\int_0^\infty \tilde{p}_T(n;\lambda,\mu,t)f(t;\lambda+\mu,\beta)\,dt$$

$$=\frac{1}{n!}\frac{\Gamma(\lambda+n)}{\Gamma(\lambda)}\frac{\Gamma(\lambda+\mu)}{\Gamma(\lambda+\mu+n)}\frac{\beta^{(\lambda+\mu)}}{\Gamma(\lambda+\mu)}\int_0^\infty {}_1F_1(\lambda+n,\lambda+\mu+n,-t)t^{\lambda+\mu+n-1}e^{-\beta t}\,dt.$$

(18)

We now apply the following identity given in *Saad and Hall, 2003* [Appendix 1] with $a=\lambda+n$, $b=\lambda+\mu+n$, $k=-1$, $d=\lambda+\mu+n$ and $h=\beta$.

$$\int_0^\infty {}_1F_1(a,b,kt)t^{d-1}e^{-ht}=\frac{\Gamma(d)}{h^d}\left(1-\frac{k}{h}\right)^{-a}.$$

(19)

So the left hand side of (18) becomes

$$\frac{1}{n!}\frac{\Gamma(\lambda+n)}{\Gamma(\lambda)}\frac{\Gamma(\lambda+\mu)}{\Gamma(\lambda+\mu+n)}\frac{\beta^{(\lambda+\mu)}}{\Gamma(\lambda+\mu)}\frac{\Gamma(\lambda+\mu+n)}{\beta^{\lambda+\mu+n}}\left(1+\frac{1}{\beta}\right)^{-(\lambda+n)}$$

$$=\frac{1}{n!}\frac{\Gamma(\lambda+n)}{\Gamma(\lambda)}\frac{\beta^\lambda}{(\beta+1)^{\lambda+n}}=\tilde{p}_{\mathrm{NB}}(n;\lambda,\beta),$$

which is the right hand side of (17). Hence, we have that (17) holds. We now consider the representation given in *Equation (5)* of the main text. Here we claim that any negative binomial distribution of the form $\mathrm{NegBin}(\lambda',\frac{b}{1+b})$ (where $b>0$) can be written as,

$$\tilde{p}_{\mathrm{NB}}(n;\lambda',b)=\int_{\frac{1}{b}}^\infty \tilde{p}_{\mathrm{NB}}(n;\lambda,\theta)f_b(b\theta-1;\lambda-\lambda',\lambda')\,d\theta,$$

(20)

where $f_b(b\theta-1;\lambda-\lambda',\lambda')$ is the probability mass function of a scaled beta prime $\mathrm{BetaPrime}_b(\lambda-\lambda',\lambda)$ distribution, where $b>0$ and $\lambda>\lambda'$. Starting from the right hand side of (20), we have

$$\int_{\frac{1}{b}}^\infty \tilde{p}_{\mathrm{NB}}(n;\lambda,\theta)f_b(b\theta-1;\lambda-\lambda',\lambda')\,d\theta$$

$$=\int_{\frac{1}{b}}^\infty \frac{\Gamma(\lambda+n)}{\Gamma(n+1)\Gamma(\lambda)}\left(1-\frac{\theta}{\theta+1}\right)^n\left(\frac{\theta}{\theta+1}\right)^\lambda\frac{b\Gamma(\lambda)}{\Gamma(\lambda-\theta)\Gamma(\theta)}\frac{(b\theta-1)^{\lambda-\lambda'-1}}{(b\theta)^\lambda}$$

$$=\frac{b^{1-\lambda}\Gamma(\lambda+n)}{\Gamma(n+1)\Gamma(\lambda')\Gamma(\lambda-\lambda')}\int_{\frac{1}{b}}^\infty \frac{(b\theta-1)^{\lambda-\lambda'-1}}{(1+\theta)^{\lambda+n}}.$$

Now substituting $H=b\theta-1$ and simplifying, we obtain

$$\frac{b^n\Gamma(\lambda+n)}{\Gamma(n+1)\Gamma(\lambda')\Gamma(\lambda-\lambda')}\int_0^\infty \frac{H^{\lambda-\lambda'-1}}{(H+b+1)^{\lambda+n}}\,dH,$$

(21)

and letting $y=b+1$ and simplifying we have that

$$\int_0^\infty \frac{H^{\lambda-\lambda'-1}}{(H+b+1)^{\lambda+n}}\,dH=\frac{1}{(b+1)^{\lambda'+n}}\int_0^\infty \frac{y^{\lambda-\lambda'-1}}{(y+1)^{\lambda+n}}\,dy$$

(22)

$$=\frac{1}{(b+1)^{\lambda'+n}}\frac{\Gamma(\lambda-\lambda')\Gamma(\lambda'+n)}{\Gamma(\lambda+n)}.$$

(23)

Here, we used the fact that the integral in the variable $y$ is the probability density function of a $\mathrm{BetaPrime}(\lambda-\lambda',\lambda'+n)$ distribution. Thus, *Equation (21)* simplifies to

$$\frac{\Gamma(\lambda'+n)}{\Gamma(n+1)\Gamma(\lambda')}\frac{b^n}{(b+1)^{\lambda'+n}}=\tilde{p}_{\mathrm{NB}}(\lambda',b),$$

(24)

which is the right hand side of (20). Thus, we have that (20) holds.

## Appendix 2

### Proof of the Noise Decomposition Principle

Here we provide the proof of the Noise Decomposition Principle given in the main text. For convenience we restate the principle below.

### The Noise Decomposition Principle (NDP)

Assume that there are measurable functions $f$, $g$ and $h$ such that $\mathrm{E}(X; \mathbf{Z})$ and $\mathrm{E}(Y; \mathbf{Z})$ split across common variables by way of $\mathrm{E}(X; \mathbf{Z}) = f(\mathbf{Z}_X)h(\mathbf{Z}')$ and $\mathrm{E}(Y; \mathbf{Z}) = g(\mathbf{Z}_Y)h(\mathbf{Z}')$. Then provided that the variables $Z_1, \ldots, Z_m$ are mutually independent, the normalised covariance of $\mathrm{E}(X; \mathbf{Z})$ and $\mathrm{E}(Y; \mathbf{Z})$ will identify the total noise on $h(\mathbf{Z}')$ (i.e., $\eta^2_{h(Z')}$).

Consider first the covariance of $\mathrm{E}(X; \mathbf{Z})$ and $(Y; \mathbf{Z})$. We have

$$
\begin{aligned}
\mathrm{Cov}(\mathrm{E}(X; \mathbf{Z}), \mathrm{E}(Y; \mathbf{Z})) &= \mathrm{Cov}(f(\mathbf{Z}_X)h(\mathbf{Z}'), g(\mathbf{Z}_Y)h(\mathbf{Z}')) \\
&= \mathrm{E}(f(\mathbf{Z_X})g(\mathbf{Z_Y})(h(\mathbf{Z}'))^2) - \mathrm{E}(f(\mathbf{Z_X})h(\mathbf{Z}'))\mathrm{E}(g(\mathbf{Z_Y})h(\mathbf{Z}')) \\
&= \mathrm{E}(f(\mathbf{Z_X}))\mathrm{E}(g(\mathbf{Z_Y}))\left[\mathrm{E}((h(\mathbf{Z}'))^2) - (\mathrm{E}(h(\mathbf{Z}')))^2\right].
\end{aligned}
\tag{25}
$$

Note that here we used the fact that the variables in $\mathbf{Z} = (Z_1, \ldots, Z_n)$ are mutually independent. Now using the fact that $\mathrm{E}(X) = \mathrm{E}(\mathrm{E}(X; \mathbf{Z}))$ and $\mathrm{E}(Y) = \mathrm{E}(\mathrm{E}(Y; \mathbf{Z}))$ (the Law of Total Expectation), and then normalising we obtain

$$
\begin{aligned}
\frac{\mathrm{Cov}(\mathrm{E}(X; \mathbf{Z}), \mathrm{E}(Y; \mathbf{Z}))}{\mathrm{E}(X)\mathrm{E}(Y)} &= \frac{\mathrm{E}(f(\mathbf{Z_X}))\mathrm{E}(g(\mathbf{Z_Y}))\left[\mathrm{E}((h(\mathbf{Z}'))^2) - (\mathrm{E}(h(\mathbf{Z}')))^2\right]}{\mathrm{E}(f(\mathbf{Z_X}))\mathrm{E}(g(\mathbf{Z_Y}))(\mathrm{E}((h(\mathbf{Z}')))^2} \\
&= \frac{\mathrm{Var}(h(\mathbf{Z}'))}{(\mathrm{E}(h(\mathbf{Z}')))^2} \\
&= \eta^2_{h(\mathbf{Z}')}.
\end{aligned}
\tag{26}
$$

Hence, under certain conditions the normalised covariance of $\mathrm{E}(X; \mathbf{Z})$ and $\mathrm{E}(Y; \mathbf{Z})$ will identify the total noise on $h(\mathbf{Z}')$.

## Appendix 3

### Pathway reporters

Justification for simulating only one copy of the gene

Our simulations and theory have been based over reporters from a single gene copy, whereas in practice there may be multiple copies of the gene that contribute to the overall mRNA. If there are mechanisms in place to distinguish the mRNA or protein of one gene copy from another then the theory and analysis we have developed in main paper holds without change. In the case where it is not possible to distinguish mRNA and protein (respectively) from the multiple gene copies, then we now observe that the general theory continues to hold, provided there is independence between the gene copies; this assumption has been verified experimentally *Skinner et al., 2016*. Here, we demonstrate this in the case of two gene copies, though the general case for more than two genes is essentially identical but is notationally cumbersome. We are considering a situation where the variable $X$ in the Noise Decomposition Principle is the sum of two independent variables $X_1, X_2$ and $Y$ is the sum of two independent variables $Y_1, Y_2$. We assume common dependence on the environmental variables $\mathbf{Z}$ so that $E(X_1; \mathbf{Z}) = E(X_2; \mathbf{Z})$, $E(Y_1; \mathbf{Z}) = E(Y_2; \mathbf{Z})$. Using these equalities and the independence of $X_1, X_2$ and $Y_1, Y_2$ in $X = X_1 + X_2$, $Y = Y_1 + Y_2$, we find the numerator of

$$\frac{\mathrm{Cov}(\mathrm{E}(X; \mathbf{Z}), \mathrm{E}(Y; \mathbf{Z}))}{\mathrm{E}(X)\mathrm{E}(Y)}$$

is simply $4\mathrm{Cov}(\mathrm{E}(X_1; \mathbf{Z}), \mathrm{E}(Y_1; \mathbf{Z}))$, while the denominator is $4\mathrm{E}(X_1)\mathrm{E}(Y_1)$. Thus the noise decomposition coincides with that for the single copy $X_1, Y_1$. Further work may be required to consider systems where there is independence between, or there is significant deviations in the gene copies.

Upper bound for the intrinsic contribution to the covariance: constitutive expression

In the main text, we claim that the error in the pathway-reporter approach in the case of mRNA-protein reporters is negligible (i.e. the error is $\ll 1$); refer to *Equation (13)* in the main text. We here provide full details of the derivation of this expression. First let $X_m$ and $X_p$ be the number of mRNA and protein produced from the same constitutive gene modelled by the 'two-stage' model, $\mathbf{M}_2$ (see *Figure 4A* (top right) of the main text). Also, let $\mathbf{Z} = \{K_m, \delta_m, K_p, \delta_p\}$.

We restate here the expression for the intrinsic contribution to the covariance of $X_m$ and $X_p$ given as *Equation (13)* of the main text.

$$\frac{\mathrm{ECov}(X_m, X_p; \mathbf{Z}))}{\mathrm{E}(X_m)\mathrm{E}(X_p)} = \frac{\alpha}{\mathrm{E}(K_m)}, \quad \text{where } \alpha = \frac{\mathrm{E}(1/(\delta_p + 1))}{\mathrm{E}(1/\delta_p)}. \tag{27}$$

We require the following expressions for the stationary mean mRNA level and protein level of the two-stage model *Thattai and van Oudenaarden, 2001*; *Singh and Hespanha, 2007*.

$$\mathrm{E}(X_m; \mathbf{Z}) = \frac{K_m}{\delta_m}, \quad \mathrm{E}(X_p; \mathbf{Z}) = \frac{K_p}{\delta_p}\frac{K_m}{\delta_m} \quad \text{and} \quad \mathrm{Cov}(X_m, X_p; \mathbf{Z}) = \frac{K_m K_p}{\delta_m(\delta_m + \delta_p)}. \tag{28}$$

Assuming that $\delta_m$ is fixed across the cell-population, and all parameters are scaled so that $\delta_m = 1$, it follows from (28), that

$$\mathrm{E}(\mathrm{Cov}(X_m, X_p; \mathbf{Z})) = \mathrm{E}\left(\frac{1}{\delta_p + 1}\right)\mathrm{E}(K_m)\mathrm{E}(K_p). \tag{29}$$

Using the Total Law of Expectation, we also have,

$$\mathrm{E}(X_m) = \mathrm{E}(K_m) \quad \text{and} \quad \mathrm{E}(X_p) = \mathrm{E}\left(\frac{1}{\delta_p}\right)\mathrm{E}(K_m)\mathrm{E}(K_p). \tag{30}$$

Thus, it follows that

$$\frac{\mathrm{E}(\mathrm{Cov}(X_m, X_p; \mathbf{Z}))}{\mathrm{E}(X_m)\mathrm{E}(X_p)} = \frac{\alpha}{\mathrm{E}(K_m)}, \quad \text{where } \alpha = \frac{\mathrm{E}(1/(\delta_p + 1))}{\mathrm{E}(1/\delta_p)}. \tag{31}$$

## Determination of the marginal means for model $\mathbf{M}_4$

In order to establish the Noise Decomposition principle in the case of bursty gene expression, we rely on expressions for the marginal stationary means of the model $\mathbf{M}_4$; see *Figure 4A* (bottom left) and the associated caption for more details. To derive the marginal means for nascent and mature mRNA and for protein (*Equation (14)* of the main text), we first observe that the nascent mRNA population may treated identically to that of mRNA in general (that is, no distinction between nascent and mature), as in *Peccoud and Ycart, 1995*, except that mRNA decay is replaced by the sum of decay and maturation. As in the work of *Cao and Grima, 2020*, the assumption of fast maturation allows us to ignore decay completely in the nascent phase, so that the marginal distribution is identical to that of *Peccoud and Ycart, 1995*, except with decay replaced by maturation. This leads to a marginal nascent mRNA mean of

$$\mathrm{E}(X_N; \mathbf{Z}) = \frac{K_N}{\delta_M} \frac{\lambda}{(\lambda + \mu)}$$

The marginal means for mature mRNA and protein are derived in *Raj et al., 2006* under the assumption that the transcription rate parameter $K_N$ is large relative to the other parameters. The expressions are given by

$$\mathrm{E}(X_M; \mathbf{Z}) = \frac{K_N}{\delta_M} \frac{\lambda}{(\lambda + \mu)} \quad \text{and} \quad \mathrm{E}(X_P; \mathbf{Z}) = \frac{K_P}{\delta_P} \frac{K_N}{\delta_M} \frac{\lambda}{(\lambda + \mu)}.$$

Formally, the marginal means in *Raj et al., 2006* are for the three-stage model $\mathbf{M}_3$, which ignores the downstream processing of mRNA, such as splicing. The assumption of fast maturation however, justifies the treatment of the nascent phase of mRNA as an ephemeral step within the Poissonian modelling of mRNA transcription.

There are a number of possibly compounding assumptions on the parameters here, but simulations show that there is a lot of tolerance, with even only moderate maturation and transcription still returning sample means consistent with the formulas.

## Measuring noise from an instantaneously bursty gene

Here we use the simple stochastic model of *Singh and Bokes, 2012* that includes both instantaneous transcriptional bursting and mRNA maturation. This model can be obtained as a limiting case of the model $\mathbf{M}_4$ (*Figure 4A* (top left) in the main text), where the off-rate $\mu$ has tended toward infinity, while the on-rate $\lambda$ remains fixed. Under this condition, transcription is rare enough to be considered instantaneous, leading to 'bursts' of transcriptional activity. The intensity of the bursts $M$ is known to be geometrically distributed with mean burst size $B = \frac{K_N}{\mu}$.

If we let $X_N$ and $X_M$ denote the number of nascent and mature mRNA, respectively, then according to *Singh and Bokes, 2012*, the steady-state marginal means and covariance are given by

$$\mathrm{E}(X_N; \mathbf{Z}) = \frac{\lambda B}{K_M}, \quad \mathrm{E}(X_M; \mathbf{Z}) = \frac{\lambda B}{\delta_M} \quad \text{and} \quad \mathrm{Cov}(X_N, X_M) = \frac{\lambda B^2}{K_M + \delta_M}. \tag{32}$$

It is easily seen from (32) that $\mathrm{E}(X_N; \mathbf{Z}) = f(\mathbf{Z}_N)B\lambda$ and $\mathrm{E}(X_M; \mathbf{Z}) = g(\mathbf{Z}_M)B\lambda$, where $f(\mathbf{Z}_N) = \frac{1}{K_M}$ and $g(\mathbf{Z}_M) = \frac{1}{\delta_M}$. Thus, the Noise Decomposition Principle holds. Using arguments similar to those given in the above section, it is straightforward to derive the following expression for the error in the pathway reporter approach:

$$\frac{\mathrm{E}(\mathrm{Cov}(X_N, X_M; \mathbf{Z}))}{\mathrm{E}(X_N)\mathrm{E}(X_M)} = \frac{\alpha}{\mathrm{E}(\lambda)}(\eta_B^2 + 1), \quad \text{where } \alpha = \frac{\mathrm{E}(1/(K_M + 1))}{\mathrm{E}(1/K_M)}. \tag{33}$$

We can expect that $\alpha \approx 1$, so that unless the burst frequency is substantial, that is $\lambda \gg 1$, the overshoot in the the nascent-mature pathway-reporter approach is significant.

## Appendix 4

### Convergence of pathway and dual reporters

**Nascent-mature reporter convergence (constitutive)**

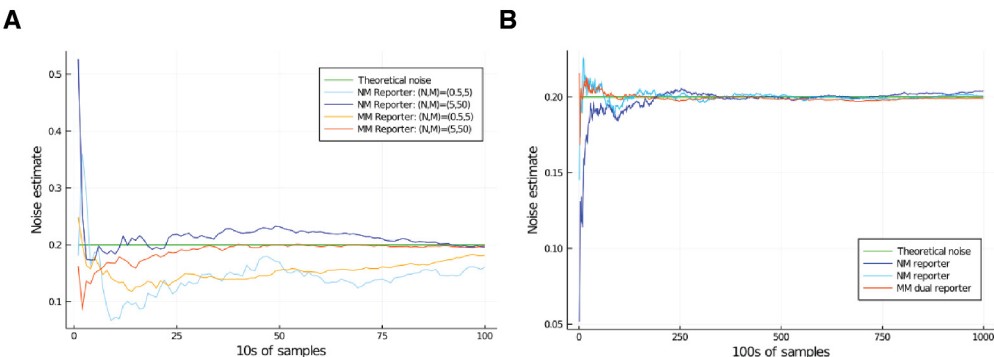

**Appendix 4—figure 1.** Comparison of convergence of $\eta^2$ estimates for low and high mRNA levels by way of nascent-mature reporters and mature-mature reporters. Low level corresponds to mean nascent mRNA level of 0.5, and mean mature mRNA level of 5. High level corresponds to mean nascent mRNA level of 5, and mean mature mRNA level of 50. In both cases, the simulated genes are constitutive and noise is on all parameters except for $\delta_M = 1$. The green line gives the squared coefficient of variation for $K_N$, set to 0.2, which is the value the various reporters are expected to estimate. (**A**) Convergence of the $\eta^2$ estimate over the first 2000 samples in the low- and high-output genes. (**B**) Convergence of the $\eta^2$ estimate over $100,000$ samples in the low-output gene only.

Convergence of both pathway reporters and dual reporters in low output genes is slower than for high output genes, and this affect is most pronounced for reporters taken from nascent mRNA. Fig. Convergence of Pathway and Dual Reporters compares convergence in some low output genes and high output genes for nascent-mature pathway reporters and the mature-mature dual reporter, in the case of constitutive expression. From *Equation (11)* of the main text and the NDP, these should measure precisely the extrinsic noise on the transcription rate parameter $K_N$. In the low output gene, both the dual reporter and pathway reporters are yet to show accurate measurement of the overall extrinsic noise after 1000 samples (Figure Convergence of Pathway and Dual ReportersA), although they are providing rough estimates after as little as a few hundred samples. Pathway reporters for nascent-mature typically exhibit slightly slower convergence than mature-mature dual reporter; the difference is marginal, but can be seen for both the high output gene (compare the values after 500 samples) and the low output gene (compare the values after 1000 samples). In this figure, all parameters (except the reference parameter $\delta_M$) were given scaled Beta distribution noise. The noise on $K_N$ is a $\mathrm{Beta}(3,6)$ distribution scaled to achieve $E(K_N) = 5$ in the low output case and $E(K_N) = 50$ in the high output case. The squared coefficient of variation is 0.2, and is shown in green in the figure.

**Mature-protein reporter convergence (constitutive)**

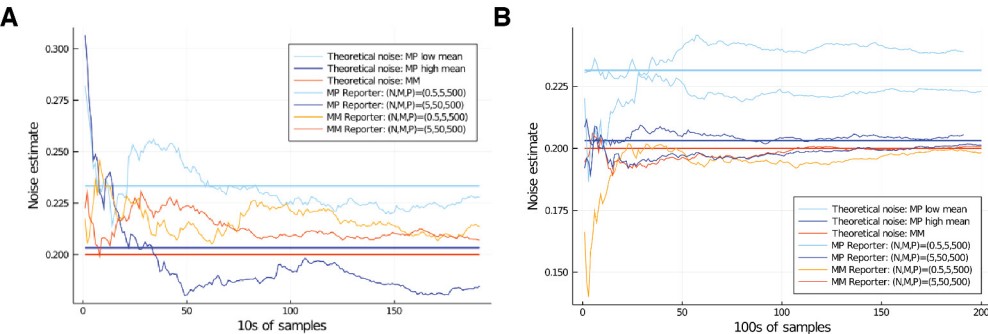

**Appendix 4—figure 2.** Comparison of convergence for low and high mRNA levels by way of mature-protein and mature-mature reporters. Low level corresponds to mean nascent mRNA level of 0.5, and mean mature mRNA level of 5. High level corresponds to mean nascent mRNA level of 5, and mean mature mRNA level of 50. In both cases the simulated genes are constitutive and noise is on all parameters except for $\delta_M = 1$. The noise on $K_N$ has squared coefficient of variation equal to 0.2, which is shown as the red horizontal line. Our theory shows that mature-protein reporters will return an overshoot that is negligible in the high-output gene (the blue horizontal line), but larger in the low output gene (light blue horizontal line); these values are calculated in the text. (**A**) Comparison of convergence for low and high mRNA levels over the first 2000 samples. (**B**) Convergence of the $\eta^2$ estimate over $20,000$ samples in the case of the low-output gene only. Two examples of each are given, to show the variation in behaviour.

Convergence for the mature-protein reporters is considered in Figure Convergence of Pathway and Dual Reporters. Here, *Equation (13)* of the main text shows that we should expect an overshoot in comparison to the mature-mature dual reporter. We have again compared a low output gene with a high output gene, and with all parameters (except $\delta_M$) experiencing noise. The transcription is again given scaled $\mathrm{Beta}(3,6)$ distribution, having squared coefficient of variation 0.2, with scaling to achieve $E(K_N) = 5$ in low output and $E(K_N) = 50$ in the high output. The noise distribution for $\delta_P$ is a $\mathrm{Beta}(8,6)$ distribution, scaled to achieve a mean value of 0.2. Computational sampling from $10^7$ samples finds the value of $\frac{\mathrm{E}(1/(\delta_p+1))}{\mathrm{E}(1/\delta_p)}$ as approximately 0.1573. The high output gene then expects an overshoot of 0.003146 from the mature-protein reporters, while the low output gene expects an overshoot of approximately 0.03146. Figure Convergence of Pathway and Dual ReportersA shows comparison of convergence over the first 2000 samples, with the theoretical values accommodating the calculated overshoot. Figure Convergence of Pathway and Dual ReportersB shows the same over 20,000 samples; this time two low-output two high-output genes are considered, so that the variation around the expected long term value can be seen. It is evident that reasonable estimates are given after a relatively modest number of samples, but there is a very long delay to a highly accurate convergence for the pathway reporters in the low output gene.

# Reporter convergence

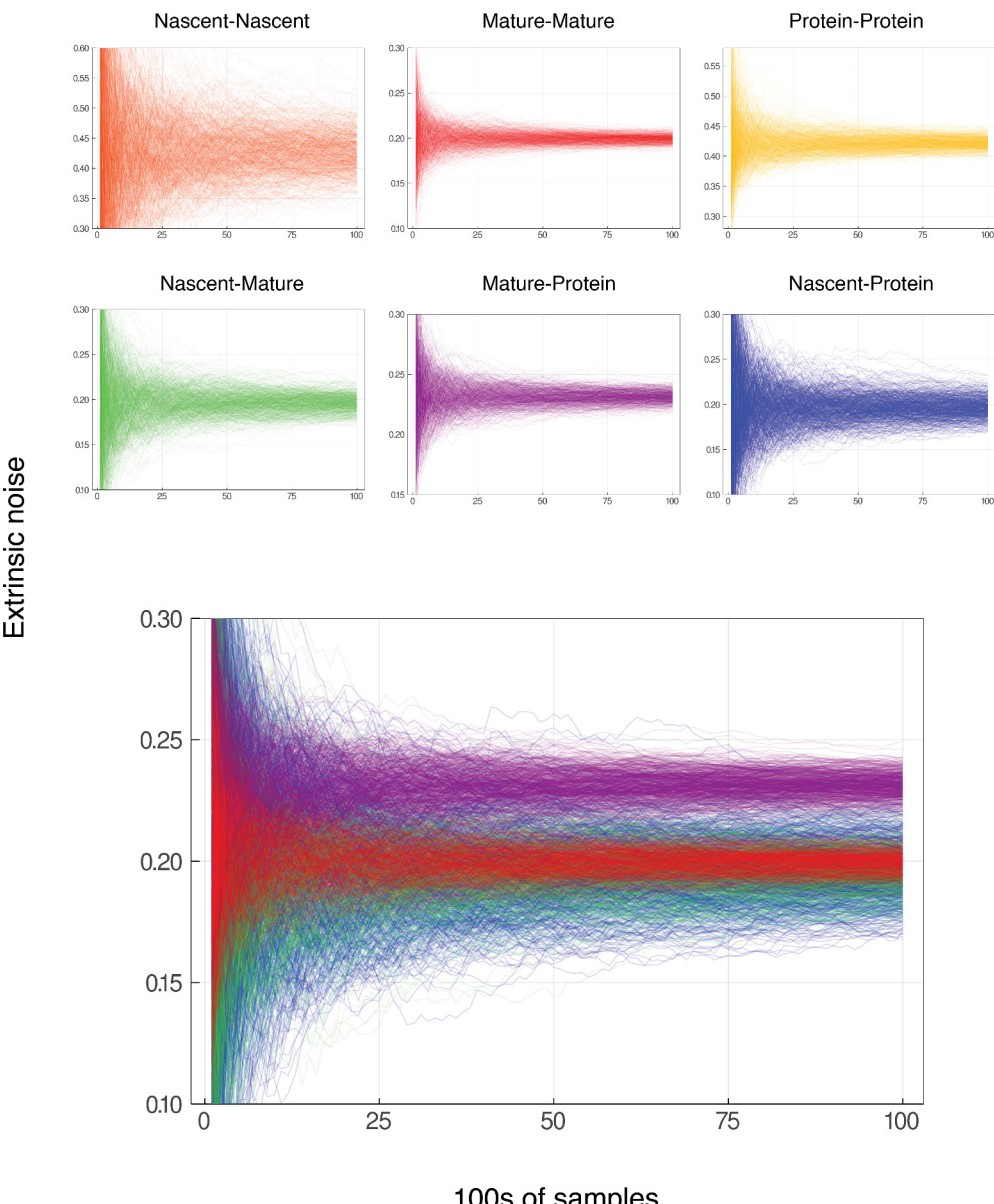

**Appendix 4—figure 3.** Convergence for reporter pairs for low gene activity. In each case, the mean nascent mRNA level is 0.5, the mean mature mRNA level of 5, and the mean protein level is 500. The simulated genes are constitutive and noise is on all parameters except for $\delta_M = 1$. Each graph shows the convergence of 600 individual reporter simulations, for each combination of reporters from nascent mRNA, mature mRNA and protein. Each reporter simulation is from $10,000$ samples, with the reporter estimates calculated at intervals of 100. The noise on $K_N$ has squared coefficient of variation equal to 0.2, which should be identified by both the nascent-mature reporter and mature-mature dual reporter. As in Figure Convergence of Pathway and Dual Reporters, the mature-protein reporter should converge to an estimate of approximately 0.2315. Nascent-nascent and protein-protein reporters identify combined noise on more parameters, so do not converge to 0.2. The lower graph shows each of nascent-mature, mature-mature, mature-protein and nascent-protein in the same plot for direct comparison.

## Appendix 5

### Generality of the pathway reporter method: model and simulation details

We verify our pathway reporter approach on synthetic data for genes with various underlying gene expression dynamics. In the main text, we consider four different scenarios, beyond the standard four-stage model of gene transcription. Below we give details of these models, and their method of simulation.

(1): First, we consider a more detailed model of the mRNA maturation process, where the nascent mRNA maturate after a more finely controlled time interval. Specifically we adjust the constitutive model of gene transcription ($\mathbf{M}_4$ of the main text with $\lambda = 1$ and $\mu = 0$) to include a fixed-duration of maturation. We also explore time-intervals sampled from Erlang distributions, to account for the fact that maturation can involve several shorter stochastic processes. The process is simulated using an adaptation of SSA, whereby the time of maturation of an individual nascent mRNA is calculated (by sample from the chosen distribution) at the point it is transcribed, and this timepoint is queued in a register of maturation times; this can be trivially adapted to other distributions; generalised-Erlang for example. At each iteration of the simulation, determination of the next event involves taking the smallest of the various exponentially sampled Markov processes (on, off, mRNA transcription, etc) and the next maturation event (the minimum of the maturation register). In the case that maturation is the next event, the corresponding maturation time is removed from the maturation register.

(2): We next consider a model of transcriptional bursting given in *Bartman et al., 2019*; *Cao et al., 2020*. In this 'multiscale' model, gene activation occurs in two steps, initially by the binding of transcription factors, and then as a secondary step by the binding and pause of the mRNA polymerase. For details of the model refer to *Figure 6* and the surrounding paragraphs in the main text. We here show that our Noise Decomposition Principle holds for all reporters taken from this gene pathway, and verify computationally that correlations between some of the reporter pairs are negligible; the process is simulated by the SSA. In the following, we assume all reactions are first-order, characterised by exponential waiting times between successive reactions. Let $X_N$ denote the number of nascent mRNA, let $X_M$ denote the number of mature mRNA, and let $X_P$ denote the number of proteins produced from the same gene. Also let $\mathbf{Z} = \{\lambda_1, \mu_1, \lambda_2, K_N, K_M, K_P, \delta_M, \delta_P\}$. We assume that the maturation rate $K_M$ is large (i.e. $K_M > \delta_M$), which is supported by experiments *Cao and Grima, 2020*. Then using the results of *Cao et al., 2020*, the stationary averages for the nascent mRNA, mature mRNA and protein levels are given by,

$$\mathrm{E}(X_N; \mathbf{Z}) = \frac{K_N}{K_M} \frac{\lambda_1 \lambda_2}{\gamma_1 \gamma_2}, \; \mathrm{E}(X_M; \mathbf{Z}) = \frac{K_N}{\delta_M} \frac{\lambda_1 \lambda_2}{\gamma_1 \gamma_2} \; \text{and} \; \mathrm{E}(X_P; \mathbf{Z}) = \frac{K_P}{\delta_P} \frac{K_N}{\delta_M} \frac{\lambda_1 \lambda_2}{\gamma_1 \gamma_2}, \tag{34}$$

respectively. Here $\gamma_1 = \lambda_1 + \mu_1$ and $\gamma_2 = K_N + \lambda_2 + \mu_1$. We begin by considering the nascent-mature pathway reporters. From (34), it is easily seen that $\mathrm{E}(X_N; \mathbf{Z}) = f(\mathbf{Z}_N) K_N \frac{\lambda_1 \lambda_2}{\gamma_1 \gamma_2}$ and $\mathrm{E}(X_M; \mathbf{Z}) = g(\mathbf{Z}_M) K_N \frac{\lambda_1 \lambda_2}{\gamma_1 \gamma_2}$, where $f(\mathbf{Z}_N) = \frac{1}{K_M}$ and $g(\mathbf{Z}_M) = \frac{1}{\delta_M}$. So the NDP holds, and the normalised covariance of $\mathrm{E}(X_N; \mathbf{Z})$ and $\mathrm{E}(X_M; \mathbf{Z})$ will identify the extrinsic noise on the transcriptional component $K_N \frac{\lambda_1 \lambda_2}{\gamma_1 \gamma_2}$. For the mature-protein reporters, we can again see from (34) that $\mathrm{E}(X_M; \mathbf{Z}) = f(\mathbf{Z}_P) \mathrm{E}(X; \mathbf{Z})$, where $f(\mathbf{Z}_P) = \frac{K_P}{\delta_P}$. Thus the NDP holds, and so the normalised covariance of $\mathrm{E}(X_M; \mathbf{Z})$ and $\mathrm{E}(X_P; \mathbf{Z})$ will identify the strength of extrinsic noise in the mRNA copy number distribution. For the nascent-protein reporters, it is easy to see that $\mathrm{E}(X_N; \mathbf{Z}) = f(\mathbf{Z}_N) K_N \frac{\lambda_1 \lambda_2}{\gamma_1 \gamma_2}$, where $f(\mathbf{Z}_N) = \frac{1}{K_M}$, and $\mathrm{E}(X_P; \mathbf{Z}) = g(\mathbf{Z}_P) K_N \frac{\lambda_1 \lambda_2}{\gamma_1 \gamma_2}$, where $g(\mathbf{Z}_P) = \frac{K_P}{\delta_M \delta_P}$. Thus, again the NDP holds, and the normalised covariance of $\mathrm{E}(X_N; \mathbf{Z})$ and $\mathrm{E}(X_P; \mathbf{Z})$ will identify the noise on the transcriptional component $K_N \frac{\lambda_1 \lambda_2}{\gamma_1 \gamma_2}$.

(3): We combine models (1) and (2) above, incorporating the fixed time maturation of (1) with the fine-level transcriptional approach of (2). Again, we employ the adaptation of the SSA described in (1).

(4): We incorporate the salient features of the cell-cycle into model (3), as well as introducing Erlang-distributed maturation times. Specifically, the model accounts for the effects of gene replication, dosage compensation, binomial partitioning of products due to cell division, as well as cell-cycle length variability. Within each phase of the cell cycle, the handling of maturation times for a

given gene copy is exactly as in Case (1) above, but within the overall system described in Case (3). To capture the cell cycle, the SSA is now further modified to handle the two key sporadic changes: gene replication and cell division. After gene replication, we perform two independent modified SSA simulations, with common parameter values. The final copy numbers are obtained as a sum of those from each gene copy. At cell division, we follow just one daughter cell, selecting the inherited copy numbers by way of binomial partitioning from the mother cell values at the point of division (including the maturation register). The length of these cell-cycle phases is sampled from the chosen Erlang distribution, on commencement of the simulation of the phase.

The results of pathway reporters for each of the Cases (1 – 4) above are given in *Appendix 5—tables 1–8* below. In all cases, the values given are the average of 20 simulations, each calculated from 500 copy number samples, and the errors are ± one standard deviation. For each model, we consider two parameter sets, each without extrinsic noise (Subcase A) and with extrinsic noise (Subcase B). For Case (1), the model parameters are chosen to produce an average nascent mRNA copy number of 10, and an average number of 2000 proteins; the average mature mRNA copy number varies according to the maturation time. We find that all of the pathway reporters accurately predict the true extrinsic noise levels (as given by dual reporters) across a range of maturation times; refer to *Appendix 5—table 1*, *2*. For Cases (2 – 4), the model parameters are chosen to produce an average nascent mRNA copy number of 5 and an average mature mRNA copy number of 100, and an average number of 2000 proteins. We verify computationally that the mature-protein and nascent-protein reporter pairs provide accurate predictions of extrinsic noise contributions across a range of noise levels. For these results refer to *Appendix 5—tables 3–8*. We mention that the results of Case (4)B (*Appendix 5—tables 8*) may suggest a slight undershoot in the pathway reporter values in comparison to dual reporters. The difference is, however, within one standard deviation, and are calculated from a relatively small sample size due to the complexity of the simulation and corresponding run time. Investigating the presence and possible causes of this marginal effect for this model and other complicated models may be of interest in further work.

**Appendix 5—table 1.** A comparison of the pathway-reporter method and dual-reporter method for constitutive gene expression with Erlang-distributed maturation times (Case (1)A).
Here, PR (NP) gives the results of the nascent and protein pathway reporters, PR (MP) gives the results of the mRNA and protein reporters, while DR (Mat) gives the results of the dual reporters calculated from the mature mRNA. The maturation time $T_{\mathrm{mat}}$ is chosen to be Erlang distributed with mean length $1/30, 0.05$, and $0.1$, respectively. We consider the rate parameters for the remaining exponentially distributed times to be constant, so that there is no extrinsic noise. The pathway-reporters correctly identify the zero extrinsic noise contribution.

| Parameters | | | | Simulation | | | |
|---|---|---|---|---|---|---|---|
| (r)1-6 $K_N$ | $T_{\mathrm{mat}}(mean)$ | $K_P$ | $\delta_P$ | Pr(NM) | Pr (MP) | Pr (NP) | DR (Mat) |
| 300 | $0.0\dot{3}$ | $0.0\dot{6}$ | 0.1 | $0.00 \pm 0.001$ | $0.00 \pm 0.0001$ | $0.00 \pm 0.0003$ | $0.00 \pm 0.0001$ |
| 200 | 0.05 | 1 | 0.1 | $0.00 \pm 0.001$ | $0.00 \pm 0.0001$ | $0.00 \pm 0.0004$ | $0.00 \pm 0.0001$ |
| 100 | 0.1 | 2 | 0.1 | $0.00 \pm 0.001$ | $0.00 \pm 0.0002$ | $0.00 \pm 0.0001$ | $0.00 \pm 0.0004$ |

**Appendix 5—table 2.** A comparison of the pathway-reporter method and dual-reporter method for constitutive gene expression and fixed maturation time (Case (1)B).
For each of the parameters $K_P, \delta_P$ we selected a scaled $\mathrm{Beta}(5,6)$ distribution, with squared coefficient of variation $\eta^2 = 0.1$; the scaling is chosen in each case to achieve a mean value equal to the parameter value. The parameter $K_N$ is given the noise distribution $\mathrm{Beta}(3,6)$, which has a slightly higher coefficient of variation $\eta^2 = 0.2$. In order to benchmark against dual reporters, the maturation time was fixed in each case. The extrinsic noise contribution predicted by the pathway-reporters matches well with the dual reporter values.

| Mean | | | | Simulation | | | |
|---|---|---|---|---|---|---|---|
| (r)1-4 $T_{\mathrm{mat}}$ | $K_N$ | $K_P$ | $\delta_P$ | Pr(NM) | Pr (MP) | Pr (NP) | DR (Mat) |

*Continued on next page*

*Appendix 5—table 2 continued*

| Mean | | | | Simulation | | | |
|---|---|---|---|---|---|---|---|
| (r)1-4 $T_{\mathrm{mat}}$ | $K_N$ | $K_P$ | $\delta_P$ | Pr(NM) | Pr (MP) | Pr (NP) | DR (Mat) |
| 0.05 | 200 | 1 | 0.1 | $0.20 \pm 0.01$ | $0.20 \pm 0.02$ | $0.20 \pm 0.03$ | $0.20 \pm 0.01$ |
| 0.1 | 100 | 2 | 0.1 | $0.20 \pm 0.01$ | $0.21 \pm 0.03$ | $0.21 \pm 0.03$ | $0.20 \pm 0.01$ |

**Appendix 5—table 3.** A comparison of the pathway-reporter method and dual-reporter method for the multiscale model (Case (2)A above).
We consider fixed parameters values (that is, no extrinsic noise). As our theory predicts, the pathway-reporters correctly identify zero extrinsic noise.

| Parameters | | | | | | Simulation | | |
|---|---|---|---|---|---|---|---|---|
| (r)1-6 $\lambda_1$ | $\mu_1$ | $\lambda_2 = K_N$ | $K_M$ | $K_P$ | $\delta_P$ | Pr (MP) | Pr (NP) | DR (Mat) |
| 2 | 2 | 400 | 20 | 2 | 0.1 | $0.02 \pm 0.003$ | $0.00 \pm 0.01$ | $0.00 \pm 0.002$ |
| 4 | 20 | 1210 | 20 | 2 | 0.1 | $0.02 \pm 0.003$ | $0.01 \pm 0.01$ | $0.00 \pm 0.01$ |

**Appendix 5—table 4.** A comparison of the pathway-reporter method and dual-reporter method for the multiscale model (Case 2.B).
For each of the parameters $\lambda_1, \mu_1, K_P, \delta_P$ we selected a scaled $\mathrm{Beta}(5,6)$ distribution, with squared coefficient of variation $\eta^2 = 0.1$; the scaling is chosen in each case to achieve a mean value equal to the parameter value. The parameter $\lambda_2 = K_N$ is given the noise distribution $\mathrm{Beta}(3,6)$, which has a slightly higher coefficient of variation $\eta^2 = 0.2$. In order to benchmark against dual reporters, the maturation rate was fixed at 20. As our theory suggests, the extrinsic noise contribution predicted by the pathway reporters matches well with the dual-reporter values.

| Mean | | | | | Simulation | | |
|---|---|---|---|---|---|---|---|
| (r)1-5 $\lambda_1$ | $\mu_1$ | $\lambda_2 = K_N$ | $K_P$ | $\delta_P$ | Pr (MP) | Pr (NP) | DR (Mat) |
| 2 | 2 | 400 | 2 | 0.1 | $0.18 \pm 0.03$ | $0.16 \pm 0.04$ | $0.16 \pm 0.02$ |
| 4 | 20 | 1210 | 2 | 0.1 | $0.29 \pm 0.04$ | $0.27 \pm 0.07$ | $0.28 \pm 0.03$ |

**Appendix 5—table 5.** A comparison of the pathway-reporter method and dual-reporter method for the multiscale model with a fixed duration of maturation (Case (3)A).
Here the time to maturation $T_{\mathrm{mat}}$ is chosen to be consistent with the mean of the stochastic maturation time used in our other models (where the maturation time is exponentially distributed). We consider all rate parameters to be constant, that is, there is no extrinsic noise. Pathway-reporters correctly identify the zero extrinsic noise contribution.

| Parameters | | | | | | Simulation | | |
|---|---|---|---|---|---|---|---|---|
| (r)1-6 $\lambda_1$ | $\mu_1$ | $\lambda_2 = K_N$ | $T_{\mathrm{mat}}$ | $K_P$ | $\delta_P$ | Pr (MP) | Pr (NP) | DR (Mat) |
| 2 | 2 | 400 | 0.05 | 2 | 0.1 | $0.02 \pm 0.004$ | $0.00 \pm 0.01$ | $0.00 \pm 0.01$ |
| 4 | 20 | 1210 | 0.05 | 2 | 0.1 | $0.02 \pm 0.003$ | $0.00 \pm 0.01$ | $0.00 \pm 0.01$ |

**Appendix 5—table 6.** A comparison of the pathway-reporter method and dual-reporter method for the multiscale model with a fixed duration of maturation (Case (3)B).
Here the maturation time, $T_{\mathrm{mat}}$, is set to 0.05. For each of the parameters $\lambda_1, \mu_1, K_P, \delta_P$, we selected a scaled $\mathrm{Beta}(5,6)$ distribution, with squared coefficient of variation $\eta^2 = 0.1$; the scaling is chosen in each case to achieve a mean value equal to the parameter value. The parameter $\lambda_2 = K_N$ is given the

noise distribution $\mathrm{Beta}(3,6)$, which has a slightly higher coefficient of variation $\eta^2 = 0.2$. The extrinsic noise values given by pathway reporters match well with those obtained by dual reporters.

| Mean | | | | | Simulation | | |
|---|---|---|---|---|---|---|---|
| (r)1-5 $\lambda_1$ | $\mu_1$ | $\lambda_2 = K_N$ | $K_P$ | $\delta_P$ | Pr (MP) | Pr (NP) | DR (Mat) |
| 2 | 2 | 400 | 2 | 0.1 | $0.18 \pm 0.02$ | $0.17 \pm 0.04$ | $0.15 \pm 0.02$ |
| 4 | 20 | 1210 | 2 | 0.1 | $0.30 \pm 0.05$ | $0.30 \pm 0.12$ | $0.28 \pm 0.05$ |

**Appendix 5—table 7.** A comparison of the pathway-reporter method and dual-reporter method for the multiscale model with Erlang-distributed maturation times and cell-cycle effects (Case (4)A).
Here, the time to maturation, $T_{\mathrm{mat}}$, is chosen to be consistent with the mean of the stochastic maturation time used in our other models (where the maturation time is exponentially distributed). Specifically, we choose $T_{\mathrm{mat}} \sim \mathrm{Erlang}(3, 60)$, with mean length $3/60 = 0.05$, matching our earlier benchmarking using exponentially-distributed maturation time, with mean length 0.05. We consider the rate parameters for the remaining exponentially distributed times to be constant, that is, there is no extrinsic noise beyond that contributed by the cell-cycle effects.

| Parameters | | | | | Simulation | | |
|---|---|---|---|---|---|---|---|
| (r)1-5 $\lambda_1$ | $\mu_1$ | $\lambda_2 = K_N$ | $K_P$ | $\delta_P$ | Pr (MP) | Pr (NP) | DR (Mat) |
| 2 | 2 | 400 | 2 | 0.1 | $0.04 \pm 0.01$ | $0.02 \pm 0.001$ | $0.04 \pm 0.01$ |
| 4 | 20 | 1210 | 2 | 0.1 | $0.02 \pm 0.003$ | $0.01 \pm 0.01$ | $0.02 \pm 0.01$ |

**Appendix 5—table 8.** A comparison of the pathway-reporter method and dual-reporter method for the multiscale model with Erlang-distributed maturation times and cell-cycle effects (Case (4)B).
The Erlang distributed maturation time is chosen as in *Appendix 5—tables 7*. For each of the parameters $\lambda_1, \mu_1, K_P, \delta_P$, we selected a scaled $\mathrm{Beta}(5, 6)$ distribution, with squared coefficient of variation $\eta^2 = 0.1$; the scaling is chosen in each case to achieve a mean value equal to the parameter value. The parameter $\lambda_2 = K_N$ is given the noise distribution $\mathrm{Beta}(3, 6)$, which has a slightly higher coefficient of variation $\eta^2 = 0.2$.

| Means | | | | | | Simulation | | |
|---|---|---|---|---|---|---|---|---|
| (r)1-6 $\lambda_1$ | $\mu_1$ | $\lambda_2 = K_N$ | $T_{\mathrm{mat}}$ | $K_P$ | $\delta_P$ | Pr (MP) | Pr (NP) | DR (Mat) |
| 2 | 2 | 400 | 0.05 | 2 | 0.1 | $0.21 \pm 0.04$ | $0.18 \pm 0.04$ | $0.23 \pm 0.02$ |
| 4 | 20 | 1210 | 0.05 | 2 | 0.1 | $0.34 \pm 0.05$ | $0.30 \pm 0.12$ | $0.36 \pm 0.02$ |

