## [Decision Letter]

**Acceptance summary:**

The revisions have substantially improved the manuscript. In particular the authors have clarified the connection of the present work to previous work and shown that their method's accuracy in identifying noise sources does not depend on the details of the gene expression model used. The novel method is based on multiple generic reporters from the same biochemical pathways. Since now it is possible to do simultaneous measurements of transcripts and proteins in single cells, this method offers a viable alternative to the standard dual reporter method, as a means to infer the magnitudes of intrinsic and extrinsic transcriptional noise.

**Decision letter after peer review:**

[Editors’ note: the authors submitted for reconsideration following the decision after peer review. What follows is the decision letter after the first round of review.]

Thank you for submitting the paper "Pathway dynamics can delineate the sources of transcriptional noise in gene expression" for consideration at *eLife*. Your article has been reviewed by 3 reviewers, one of whom is a member of our Board of Reviewing Editors, and the evaluation has been overseen by a Senior Editor. Although the work is of interest, we are not convinced that the findings presented have the potential significance that we require for publication in *eLife*.

Specifically, some of the main critical comments overlapping between one or more reviews are: the need for experimental confirmation of the approach, major issues with the model assumptions and potential overlap with previously published works. Several recommendations have been made by the referees and we hope you find these useful. However it appears that the changes needed are considerable and the paper would need to be rewritten, hence the decision to reject the paper in its current form.

*Reviewer #1:*

Ham et al. present a manuscript that attempts to answer two main questions:

(i) Is it possible to identify the relative sources of population heterogeneity from

the measured transcript abundance distribution alone?

(ii) Can one develop a method that can reliably estimate the strength of intrinsic and extrinsic noise and that does not require identical and independent pairs of

gene reporters?

In the context of this paper, extrinsic noise is assumed to be variations in parameters across cells; these variations are static and do not depend on time. The authors then proceed to answer question (i) by writing the observed distribution of transcripts as a compound distribution given by Equation 2 which takes as input the distribution of the telegraph model (that is analytically known in closed-form) – the underlying dynamics – and the distribution of parameters across a cellular population – the population heterogeneity. As summarized in Table I, they find that there are various different choices of these 2 distributions that can lead to the same compound (observed) distribution, thus highlighting the issue of non-identifiability if one only has available transcription abundance distributions. I note that the issue of non-identifiability is already known and described in the literature, however the authors identify some cases that were previously unknown. This framework is useful and the results are interesting; however the present formalism does not take into account some of the most important and ubiquitous sources of noise, namely those due to cell division (binomial partitioning of products upon division) and the cell-cycle (replication and cell-cycle length variability) because the authors use the conventional telegraph model that does not account for these phenomena – see Ref 38 where it is shown that these sources of noise can easily mimic that due to transcriptional bursting.

In the second part of the paper, the authors address question (ii). The dual reporter method of Swain et al. (Ref 4) is the standard method to decompose noise into its intrinsic and extrinsic contributions by using two independent reporter genes integrated into the same cell. The assumption that the reporters have identical dynamics is a strong one and presents difficulties in interpreting the results. Ham et al. present an alternative method which does not need dual reporters, rather makes use of 2 pieces of data (for e.g. mRNA and protein) belonging to the same biochemical pathway. The result is based on the decomposition of the covariance of any two variables according to the law of total covariance. This "pathway-reporter method" is shown to be accurate provided there is a small correlation between the reporter pairs – this is a strong assumption, in my view, and it is difficult to make a strong case for it generally. Also when carrying out verification of the method using synthetic data, the correlations between reporter pairs may strongly depend on the details of the model used. For e.g. nascent mRNA maturation to mRNA is in the present paper modeled via a one-step first-order process but more biologically faithful models (see for example PMID: 27667861 and 33976195) model this via a reaction step with a fixed time delay (modelling elongation + termination). The correlations calculated using the latter are likely stronger than the former because the former assumes exponentially distributed times. My main concern however is that the method presented is similar to another one described in the paper PMID: 22529351 which also uses the law of total covariance to understand how the variation in a species is determined by other species in the same pathway. There is also no comparison of the present method with another common one whereby the extrinsic noise is estimated as the part of the expression for the coefficient of variation that is not dependent on the mean molecule numbers; see for e.g. Ref 38 and Ref. 55 (Figure 2B).

*Reviewer #1:*

I found this paper very interesting to read and I think there is ample scope for the development of new methods in this field. Hence I am generally supportive of this paper.

My main concerns that I would like the authors to address, in order of importance, are:

(i) Novelty – is their method different than PMID: 22529351? They seem to be based on similar mathematical formalisms.

(ii) Nascent mRNA is more faithfully modeled using a delay step rather than a first-order step. Using the latter is approximative at best but will certainly decrease the correlations and maybe making your method perform pathway-reporter better than it would otherwise. The issue might be that the inclusion of such a delay step will lead to a non-Markovian model in which case I am not sure that the same mathematical steps follow

(iii) Cell-cycle and cell division effects are strong sources of noise and it would be ideal that these are at least discussed extensively; even better if they can be integrated into a modified telegraph model and then one can ask the same questions about what one can identify from a compound distribution based on Equation 2.

(iv) Discussion of their method vs the other method of obtaining the extrinsic noise from plots of the coefficient of variation squared versus molecule numbers.

(iv) The manuscript should be checked carefully as I found a number of grammatical mistakes scattered throughout.

*Reviewer #2:*

The paper deals, theoretically, with the question of decomposing noise into intrinsic and extrinsic components. This has previously been done primarily using the dual reporter method. Here, the authors suggest alternative methods that could in principle bypass the need for using two reporters.

I do not find the results very relevant from the biological perspective. The authors only verify their methods on stochastic simulations, where the model assumptions are specified by hand and thus fit perfectly into their theoretical framework. But how the proposed protocols would perform on experimental data remains unclear. The authors also seem unaware of recent work where essentially the same problem is addressed (Lin and Amir, PRL 2021) – inferring extrinsic vs. intrinsic noise from data – and a method not relying on dual reporters is tested on two existing experimental datasets. Related to the above, it was unclear to me precisely what sort of data would be needed in practice to implement their pathway-reporter method – and whether such data are currently available.

Another example of the potential discrepancy between mathematics and working with actual data is the authors' discussion of the identifiability problem. It seems to me that in reality distributions are always subject to noise (e.g. due to sampling errors) and it is unclear to me whether the sort of rigorous analysis they perform in the paper (that shows whether or not identifiability is possible or not in principle, assuming a perfectly measured distribution) is relevant at all to real data. (e.g.. even if two distributions are mathematically different, would it be possible to distinguish them in practice?)

It seems to me that the paper in its current format is a much better fit for specialized journals in biophysics/mathematics such as the biophysical journal or physical review.

*Reviewer #3:*

It is now well-known that single-cell expression data exhibits significant cell-to-cell heterogeneity, which can stem from both "extrinsic" factors (e.g. enzyme concentrations, energy content, cellular environment etc.) or "intrinsic" noise (e.g. firing of reactions). Ham et al. rigorously show how extrinsic factors can cause non-identifiability of a model from single time-point gene-expression data, and also lead to incorrect conclusions about the underlying process if they are excluded from the model. Then they propose a novel pathway reporter (PR) approach for quantifying the contributions of the extrinsic and the intrinsic factors to the overall cell-to-cell heterogeneity. These results are illustrated with examples based on the telegraph gene-expression model which includes both nascent and mature mRNA. While the developed results are interesting and mathematically accurate, they come with many caveats, and they do not present a significant advance over existing works on this topic. My reasons for this assessment are mentioned below:

1. Non-identifiability results: The observations regarding non-identifiability of the dynamic parameters from the compound distribution model are not surprising and even though specific examples in Table 1 are nicely worked out, I do not see how they contribute substantially to the existing knowledge on the subject. Essentially non-identifiability will hold for nearly all choices of tilde{p} and f, and identifiability, if it exists, is an exception rather than the norm. This is consistent with what the authors observed in the simulation study outlined at the end of page 10.

2. The Pathway Reporter (PR) scheme: The PR scheme is presented as an alternative to the dual reporter (DR) scheme. However PR also suffers from issues similar to the DR scheme:

– DR assumes conditional independence of reporters. Likewise, PR also requires that such an independence (at least at the level of covariance) in order to argue that the intrinsic noise contribution is zero in the reporter covariance. The paper establishes this approximately for certain parameter regimes by using analytical expressions for the steady-state covariances. However generally such neat expressions are not readily available. It is unclear how one can check that this expected covariance is small. Of course if a fully identified model is available, then one can check it with simulations, but then one can simply compute the extrinsic noise directly.

– Another critical assumption of the PR scheme is that the conditional expectation for the reporters given the extrinsic factors Z, should nicely separate into a function of the common parameters and a function of independent parameters. Even though this works for the specific linear gene-expression model considered in the paper, it is unclear if this would work more generally. Moreover having such a splitting is (up to an additional scaling factor) is not much more general than having identical conditional expectations, as needed by the DR approach.

3. Applicability to experimental studies: As the authors mention in the paper, it is difficult to experimentally construct reporters satisfying the conditions of the DR scheme. However the paper does not discuss how for unknown experimental systems one can identify/construct reporters that satisfy the conditions of the PR scheme. In light of my previous comment, this seems to be as challenging as the DR approach, if one does not have a well-characterized mathematical model.

4. Lack of a meaningful connection: There is very little connection between the two parts of the paper (non-identifiability and PR scheme). A more substantial connection is expected, as in the abstract it is said that: "Here we mathematically formalize this non-identifiability; but we also use this to identify how new experimental set-ups coupled to statistical noise decomposition can resolve this non-identifiability."

An example showing how noise decomposition helped turn a non-identifiable system into an identifiable one is lacking. Moreover, it is unclear how quantification of the extrinsic noise enables resolution of the challenging non-identifiability issue in the compound model.

5. All the results are presented for a specific gene-expression model and its derivatives. As the paper claims to provide a general approach for analyzing noise sources, more examples need to be provided, for a clearer evaluation of the PR method and comparison with the dual reporter method.

6. The paper only considers single time-point snapshot data measured at steady-state. What happens if multiple time-points, including the transience, is also taken into account. Certainly model identifiability would improve and the PR scheme should still work. The authors should consider such examples in the paper.

7. The description of the dual reporter method is misleading – at several places the paper says that the dual reporter method requires "independent gene-reporters". However only conditional independence is required by the method. If there is full independence, then the covariance (extrinsic noise) would always be zero.

8. On page 21, first paragraph it is mentioned that "For nascent-protein reporters, the normalized intrinsic contribution to the covariance is satisfactorily small (less than 0.05) for (a) high values of δp in unison with (b) low values of λ (less than 1, though lower values are acceptable if δp is small)". However based on the discussion prior/post this statement and also Figure 5, it seems that λ values should be high (not low!) for the normalized intrinsic contribution to the covariance to be small.

---

## [Author Response]

[Editors’ note: The authors appealed the original decision. What follows is the authors’ response to the first round of review.]

Reviewer #1:[…] I found this paper very interesting to read and I think there is ample scope for the development of new methods in this field. Hence I am generally supportive of this paper.My main concerns that I would like the authors to address, in order of importance, are:(i) Novelty – is their method different than PMID: 22529351? They seem to be based on similar mathematical formalisms.

These are based on similar mathematical formalisms, but beyond this there is no appreciable overlap. To clarify, the primary similarity is that both papers involve an analysis of the covariance between reporters. In our submission, this analysis is between two reporters in the same pathway. In the article PMID: 2259351, this is between reporters in different genes. While it is true that PMID: 2259351 considers dual reporters (*not* pathway reporters) from multiple levels of a system, we emphasise that this is not the same as an analysis of covariance between single reporters from multiple levels of a system, as we do. There is certainly relevance of the PMID: 2259351 method in our broader discussion of noise decomposition, and we have now included this in the revised manuscript.

(ii) Nascent mRNA is more faithfully modeled using a delay step rather than a first-order step. Using the latter is approximative at best but will certainly decrease the correlations and maybe making your method perform pathway-reporter better than it would otherwise. The issue might be that the inclusion of such a delay step will lead to a non-Markovian model in which case I am not sure that the same mathematical steps follow

While we understand that this is not the primary concern of the referee, we agree that it is a point deserving consideration. We would first like to address the validity of the model for nascent mRNA maturation used in the submitted manuscript. This model is standard in the literature and the article PMID: 27667861 states that the model we use is a “reasonable interpretation of the release kinetics" of the maturation process. PMID: 27667861 also notes that the release kinetics are currently not well-understood but that the nascent mRNA distribution is insensitive to whether the maturation time is exponentially distributed or is modelled as a deterministic time delay, provided that the transcription rate is significantly greater than the switching (on/off rates) rates; this assumption is generally the case experimentally (refer to Section 14.3 and Figure S3 of the Supplementary Material). Our expectation then is that the nascent/mature-mRNA pathway reporter scheme will produce similar results irrespective of the details of the model used.

In order to substantiate this intuition, we have now tested the performance of our method for an alternative model of maturation, where the nascent mRNA maturate after a fixed amount of time. We also explore maturation times sampled from Erlang distributions to account for the fact that maturation can involve several shorter stochastic processes. As expected, we find that pathway reporters continue to correctly identify the extrinsic noise contribution across a range of maturation times; a discussion of these results have now been added to the revised manuscript (see the new section titled “Generality of the Pathway Reporter method"). We mention, in addition, that the issue does not arise for the mature-mRNA/protein (or the nascent-mRNA/protein) pathway reporter method in our article, which for at least some reasonable parameter ranges, is sufficient to provide good estimates of extrinsic noise.

Furthermore, we have now additionally considered various other gene expression dynamics, including a nuanced model of transcriptional bursting (capturing polymerase recruitment and pause release), as well as models that account for cell-cycle effects such as gene replication, dosage compensation, binomial partitioning due to cell division, and cell-cycle length variability. In all cases, the pathway reporter method is able to accurately identify noise sources, showing that our approach does not depend on the precise dynamics of mRNA and protein synthesis. In particular, we find that the correlations between pathway reporter pairs do not depend on the details of the gene expression model used. A new section that encompasses these results, as well as addressing the generality of the pathway reporter method has been added to the revised manuscript; please refer to the new section titled “Generality of the Pathway Reporter method" and the supplementary material. Together our results continue to suggest that our method will serve as a useful tool for gene expression analysis.

(iii) Cell-cycle and cell division effects are strong sources of noise and it would be ideal that these are at least discussed extensively; even better if they can be integrated into a modified telegraph model and then one can ask the same questions about what one can identify from a compound distribution based on Equation 2.

Cell-cycle and cell division/replication effects do indeed have a significant impact on gene expression variability. Cell-cycle effects arise predominantly from differences in the transcription rate across the cell population, and can be modelled already by Equation 2 under reasonable assumptions, without modification to the underlying telegraph model. A more explicit treatment of the mechanisms and changes along the cell cycle are challenging to study analytically (see Reference 38 for example), and theoretical models to account for such details are only starting to emerge. However, we agree that an interesting future direction would be to address how taking into account such details in the context of Equation 2 may also affect the inference of biochemical parameters. At this stage, we feel that a more detailed discussion of these effects should be included, and we have now amended the manuscript to reflect this; please refer to the final paragraph of the section titled “The Compound Distribution".

(iv) Discussion of their method vs the other method of obtaining the extrinsic noise from plots of the coefficient of variation squared versus molecule numbers.

The method of obtaining extrinsic noise from such plots is mentioned already in the introduction of the submitted manuscript. Recent work of two of the authors proves that this method is subject to non-identifiability issues (at least from measurements of transcript abundance); see Ham *et al.*, Phys. Rev. Lett, 2020 (Reference 12 of the submitted manuscript). There it is demonstrated that the coefficient of variation squared as a function of copy numbers cannot distinguish the presence of extrinsic noise.

(iv) The manuscript should be checked carefully as I found a number of grammatical mistakes scattered throughout.

The manuscript has been thoroughly checked, and all grammatical mistakes that we can find have been amended.

Reviewer #2:The paper deals, theoretically, with the question of decomposing noise into intrinsic and extrinsic components. This has previously been done primarily using the dual reporter method. Here, the authors suggest alternative methods that could in principle bypass the need for using two reporters.I do not find the results very relevant from the biological perspective. The authors only verify their methods on stochastic simulations, where the model assumptions are specified by hand and thus fit perfectly into their theoretical framework. But how the proposed protocols would perform on experimental data remains unclear.

This is a very valid observation in general concerning the translation from mathematical models to practice. The article considers already some existing databases of transcriptomic data, finding that around 90% of the genes lie within the parameter range for reasonable estimates via our method (and certainly identification of the presence of extrinsic noise), with 70% in the range for accurate measurement. We also give discussion of newly emerging experimental approaches required for the most widely applicable of our pathway combinations; these are in the final discussion of the article.

While for some models we have been able to mathematically prove that our method applies, we have now considered the robustness of our pathway reporter method on more nuanced models of gene transcription. More specifically, we consider models incorporating complex maturation processes (at the response of Referee #1), nuanced models of transcriptional bursting (capturing polymerase recruitment and pause release), as well as incorporating the effects of the cell cycle such as gene replication, dosage compensation, binomial partitioning of products due to cell division and cell-cycle length variability. In all cases, the pathway reporter method is able to accurately identify noise sources, showing that our approach does not depend on the precise dynamics of mRNA and protein synthesis. We hope this will help in assuring the referee that the method does not significantly depend on the precise details of the gene expression dynamics, nor on the availability of analytical formulas. A new section that encompasses these results has been added to the revised manuscript; please refer to the section titled “Generality of the Pathway Reporter Method" in the revised version.

The authors also seem unaware of recent work where essentially the same problem is addressed (Lin and Amir, PRL 2021) – inferring extrinsic vs. intrinsic noise from data – and a method not relying on dual reporters is tested on two existing experimental datasets. Related to the above, it was unclear to me precisely what sort of data would be needed in practice to implement their pathway-reporter method – and whether such data are currently available.

We thank the reviewer for drawing our attention to this work. The article by Lin and Amir was yet to appear at the time of submission, which partly explains why the work slipped the attention of the authors. Lin and Amir’s work does offer an alternative approach to decomposing noise, which we now reference in the revised manuscript. The generality of their approach appears comparable to ours, and is demonstrated on what is now a similar range of model variations. The applicability also appears comparable: their approach relies upon lineage (single-cell trajectory) measurements of protein concentration over many generations, and such high throughput measurements have only recently become available. Some of our reporter pairs also require experimental data that is only recently emerging, while others are obtainable from nascent/mature-mRNA measurements, which have been routinely achieved for some time.

Another example of the potential discrepancy between mathematics and working with actual data is the authors' discussion of the identifiability problem. It seems to me that in reality distributions are always subject to noise (e.g. due to sampling errors) and it is unclear to me whether the sort of rigorous analysis they perform in the paper (that shows whether or not identifiability is possible or not in principle, assuming a perfectly measured distribution) is relevant at all to real data. (e.g.. even if two distributions are mathematically different, would it be possible to distinguish them in practice?)

We respectfully disagree that mathematically different distributions in general will not be distinguishable in practice: the calibration of models to data is one of the central pillars of the scientific method, and in gene expression in particular, has lead to significant improvements in the understanding of the underlying mechanisms. The non-identifiability results presented in the manuscript show that the concern expressed by the referee *does* indeed hold in the case of the analysis of transcript abundance alone. It is a particularly strong instance, as it shows that totally different processes can present identically in experimental data. This is proved using precise distributions, but the manuscript verifies that the effect is robust, and that “for practical purposes any similar distribution will produce a similar effect”; see Figure 2B for example. A direct consequence of our results is that extrinsic noise invariably leads to data appearing more bursty than the true underlying system; we prove this mathematically for the most popular model underlying current gene expression studies, and demonstrate this through simulations for other instances. Discussion of this important observation can be found in the introductory section as well as in the section just mentioned. To ensure that these points are made as clearly as possible, we have now amended some of the phrasing in the manuscript.

It seems to me that the paper in its current format is a much better fit for specialized journals in biophysics/mathematics such as the biophysical journal or physical review.Reviewer #3:It is now well-known that single-cell expression data exhibits significant cell-to-cell heterogeneity, which can stem from both "extrinsic" factors (e.g. enzyme concentrations, energy content, cellular environment etc.) or "intrinsic" noise (e.g. firing of reactions). Ham et al. rigorously show how extrinsic factors can cause non-identifiability of a model from single time-point gene-expression data, and also lead to incorrect conclusions about the underlying process if they are excluded from the model. Then they propose a novel pathway reporter (PR) approach for quantifying the contributions of the extrinsic and the intrinsic factors to the overall cell-to-cell heterogeneity. These results are illustrated with examples based on the telegraph gene-expression model which includes both nascent and mature mRNA. While the developed results are interesting and mathematically accurate, they come with many caveats, and they do not present a significant advance over existing works on this topic. My reasons for this assessment are mentioned below:1. Non-identifiability results: The observations regarding non-identifiability of the dynamic parameters from the compound distribution model are not surprising and even though specific examples in Table 1 are nicely worked out, I do not see how they contribute substantially to the existing knowledge on the subject. Essentially non-identifiability will hold for nearly all choices of tilde{p} and f, and identifiability, if it exists, is an exception rather than the norm. This is consistent with what the authors observed in the simulation study outlined at the end of page 10.

We agree that our non-identifiability results may not be surprising to some audiences (non-identifiability issues have already been reported empirically), and we are open about this in the manuscript. This does not, however, undermine the value in proving that they are true and uncovering previously unknown, and more encompassing, non-identifiability cases. One of the results, for example, has recently formed the basis of the research paper doi.org/10.1101/2020.09.25.312868.

Our non-identifiability results are in many ways more of an entry point to our main noise decomposition methods. However, there is a very important consequence of this preliminary section: (as noted in the comments to Referee #2) the results demonstrate that extrinsic noise invariably leads to data appearing more bursty than the actual underlying system, and that this holds for all parameter regimes of the standard models. We feel this is a particularly important message to be highlighted, especially to an interdisciplinary community where the analysis of transcriptional processes increasingly relies on calibrating models to experimental data.

2. The Pathway Reporter (PR) scheme: The PR scheme is presented as an alternative to the dual reporter (DR) scheme. However PR also suffers from issues similar to the DR scheme:– DR assumes conditional independence of reporters. Likewise, PR also requires that such an independence (at least at the level of covariance) in order to argue that the intrinsic noise contribution is zero in the reporter covariance. The paper establishes this approximately for certain parameter regimes by using analytical expressions for the steady-state covariances. However generally such neat expressions are not readily available. It is unclear how one can check that this expected covariance is small. Of course if a fully identified model is available, then one can check it with simulations, but then one can simply compute the extrinsic noise directly.

While we used the specific expressions to prove approximate independence (conditional on the cell environment), the approximate independence can also be verified across a range of models, where analytical expressions are not available. Our computational efforts already provided this corroboration to some extent, and we have now explored a broader range of models, with the same result. Please refer to our response of point (5) below for more details of these newly added models.

– Another critical assumption of the PR scheme is that the conditional expectation for the reporters given the extrinsic factors Z, should nicely separate into a function of the common parameters and a function of independent parameters. Even though this works for the specific linear gene-expression model considered in the paper, it is unclear if this would work more generally. Moreover having such a splitting is (up to an additional scaling factor) is not much more general than having identical conditional expectations, as needed by the DR approach.

The Noise Decomposition Principle presented in the paper is a convenient and powerful framework that appears to be general enough to capture a flexible range of reporter pairs to estimate extrinsic noise. The assumptions are not restrictive. We have focused on the linear gene expression model because it is broadly applicable, and we can use the available analytical formulas to provide a mathematical verification of the method. As in the previous response, however, the computational simulations also provide a demonstration of what our method is capable of measuring. We have now tested the theoretical framework in extensive simulations for a range of other, more nuanced, gene expression dynamics, to explore applicability in the absence of analytical solutions. For these results, refer to the new section in the manuscript titled "Generality of the Pathway Reporter Method". The results show that our approach is remarkably insensitive to the specific dynamics of mRNA and protein synthesis, and therefore should be broadly applicable to a range of experimental scenarios.

3. Applicability to experimental studies: As the authors mention in the paper, it is difficult to experimentally construct reporters satisfying the conditions of the DR scheme. However the paper does not discuss how for unknown experimental systems one can identify/construct reporters that satisfy the conditions of the PR scheme. In light of my previous comment, this seems to be as challenging as the DR approach, if one does not have a well-characterized mathematical model.

As discussed, the general approach is verified computationally across a range of models, including those where we cannot rely on precise formulas. Additionally, the models we consider are widely used for identifying modes of gene activity from experimental data. In such situations, our assumptions have been implicitly invoked already, and the consequences of our observations would appear to hold the same validity as other consequences frequently being drawn from experimental data.

4. Lack of a meaningful connection: There is very little connection between the two parts of the paper (non-identifiability and PR scheme). A more substantial connection is expected, as in the abstract it is said that: "Here we mathematically formalize this non-identifiability; but we also use this to identify how new experimental set-ups coupled to statistical noise decomposition can resolve this non-identifiability."An example showing how noise decomposition helped turn a non-identifiable system into an identifiable one is lacking. Moreover, it is unclear how quantification of the extrinsic noise enables resolution of the challenging non-identifiability issue in the compound model.

We initially show that mRNA measurements alone cannot be used to identify even the presence of extrinsic noise; we then proceed to show that with pathway reporters it can; and moreover, we can measure its strength. We feel that this is a very clear connection. The pathway reporter scheme resolves non-identifiability to the extent that the presence of extrinsic noise can be identified and measured, whereas from copy number data alone it cannot, as we demonstrate in the first part of the manuscript. There is no claim in the article that we can identify the precise noise distribution in a compound model, and we are not sure this can be done in general, though further consideration of that claim will be an interesting topic for future work. It is, we believe, important to be able to demonstrate convincingly and unequivocally the presence of extrinsic noise from experimental data.

5. All the results are presented for a specific gene-expression model and its derivatives. As the paper claims to provide a general approach for analyzing noise sources, more examples need to be provided, for a clearer evaluation of the PR method and comparison with the dual reporter method.

We thank the reviewer for this suggestion, as we believe the addition of a more robust evaluation of the pathway reporter method has substantially improved the paper. We have now considered various other gene expression dynamics, including detailed models of the maturation process (at the suggestion of Referee #1), a more nuanced model of transcriptional bursting (capturing polymerase recruitment and pause release), as well as accounting for the cell-cycle effects such as gene replication, dosage compensation, binomial partitioning of products due to cell division, and cell-cycle length variability. In all cases, the pathway reporter method is able to accurately identify noise sources, showing that our approach does not depend on the precise dynamics of mRNA and protein synthesis. In particular, we find that the correlations between pathway reporter pairs do not depend on the details of the gene expression model used. A new section that encompasses these results, as well as addressing the generality of the pathway reporter method has been added to the revised manuscript. Together our results continue to suggest that our method will serve as a useful tool for gene expression analysis.

6. The paper only considers single time-point snapshot data measured at steady-state. What happens if multiple time-points, including the transience, is also taken into account. Certainly model identifiability would improve and the PR scheme should still work. The authors should consider such examples in the paper.

The extension to multiple time points is certainly of interest to us. In previous and related work, we have considered time course data for cells and the identifiability of model characteristics (PMID:21551095) in contrast to snapshot data. So this is definitely an avenue for further exploration that we intend to follow. But in the submitted manuscript, the focus is on the development of an alternative noise decomposition for snapshot data as such data sets are becoming readily available. The analysis of the transient behaviour of dissipative dynamical systems in the presence of noise requires a much more detailed analysis than we attempt here and is clearly outside the scope of what can be dealt with here.

7. The description of the dual reporter method is misleading – at several places the paper says that the dual reporter method requires "independent gene-reporters". However only conditional independence is required by the method. If there is full independence, then the covariance (extrinsic noise) would always be zero.

We have now amended the manuscript to make this implicit understanding explicit.

8. On page 21, first paragraph it is mentioned that "For nascent-protein reporters, the normalized intrinsic contribution to the covariance is satisfactorily small (less than 0.05) for (a) high values of δp in unison with (b) low values of λ (less than 1, though lower values are acceptable if δp is small)". However based on the discussion prior/post this statement and also Figure 5, it seems that λ values should be high (not low!) for the normalized intrinsic contribution to the covariance to be small.

We thank the reviewer for drawing our attention to this. It appears that this error was introduced at some stage in preparation (it is expressed exactly as the referee suggests in the original BioaRχiv version of the manuscript, for example). We regret this potential cause for confusion, and have carefully checked and corrected all typographical mistakes that we can find in the paper.